# Meioc-Piwil1 complexes regulate rRNA transcription for differentiation of spermatogonial stem cells

Toshihiro Kawasaki[1,2]*, Toshiya Nishimura[3], Naoki Tani[4], Carina Ramos[5], Emil Karaulanov[6], Minori Shinya[1,2], Kenji Saito[1], Emily Taylor[5], René F Ketting[6], Kei-ichiro Ishiguro[4], Minoru Tanaka[3], Kellee R Siegfried[5], Noriyoshi Sakai[1,2]*

[1]Department of Gene Function and Phenomics, National Institute of Genetics, Mishima, Japan; [2]Department of Genetics, School of Life Science, SOKENDAI (The Graduate University for Advanced Studies), Mishima, Japan; [3]Graduate School of Science, Nagoya University, Nagoya, Japan; [4]Institute of Molecular Embryology and Genetics, Kumamoto University, Kumamoto, Japan; [5]Biology Department, University of Massachusetts Boston, Boston, United States; [6]Institute of Molecular Biology (IMB), Mainz, Germany

*For correspondence:
tokawasa@nig.ac.jp (TK);
nosakai@nig.ac.jp (NS)

Competing interest: The authors declare that no competing interests exist.

## eLife Assessment

This **important** paper describes the regulatory pathway of rRNA synthesis by Meioc-Piwil1 in germ cell differentiation in zebrafish. Using the molecular genetic and cytological approaches, the authors provide **convincing** evidence that Meioc antagonizes Piwil1, which downregulates the 45S pre-rRNA synthesis by heterochromatin formation for spermatocyte differentiation. The results will be of use to researchers in the field of germ cell/meiosis as well as RNA biosynthesis and chromatin.

**Abstract** Ribosome biogenesis is vital for sustaining stem cell properties, yet its regulatory mechanisms are obscure. Herein, we show unique properties of zebrafish *meioc* mutants in which spermatogonial stem cells (SSCs) do not differentiate or upregulate rRNAs. Meioc colocalized with Piwil1 in perinuclear germ granules, but Meioc depletion resulted in Piwil1 accumulation in nucleoli. Nucleolar Piwil1 interacted with 45S pre-rRNA. *piwil1*[+/-] spermatogonia with reduced Piwil1 upregulated rRNAs, and *piwil1*[+/-];*meioc*[-/-] spermatogonia recovered differentiation later than those in *meioc*[-/-]. Further, Piwil1 interacted with Setdb1 and HP1α, and *meioc*[-/-] spermatogonia exhibited high levels of H3K9me3 and methylated CpG in the 45S-rDNA region. These results indicate that zebrafish SSCs maintain low levels of rRNA transcription with repressive marks similar to *Drosophila* piRNA targets of RNA polymerase II, and that Meioc has a unique function on preventing localization of Piwil1 in nucleoli to upregulate rRNA transcripts and to promote SSC differentiation.

## Introduction

The maintenance and control of the differentiation of adult stem cells are indispensable for living animals. In this decade, the relationship between the stem cell system and translation activity control has been revealed. Compared with that of differentiating progenitor cells, the global translation activity is relatively lowered in most adult and embryonic stem (ES) cells, and low translation activity is required for stem cells to maintain undifferentiated status (*Ni and Buszczak, 2023*; *Saba et al., 2021*). Despite this common feature of low translational activity, differences in the state of ribosome

biogenesis (RiBi) have been observed across stem cell types. ES cells, hair follicle stem cells, and *Drosophila* germline stem cells (GSCs) are characterized by high levels of RiBi, whereas quiescent neural stem cells (NSCs) have low levels of RiBi (*Llorens-Bobadilla et al., 2015*). Hematopoietic stem cells (HSCs) have higher RiBi levels than differentiated bone marrow cells, but they have lower RiBi levels than their immediate progenitors, suggesting that cell type-specific mechanisms may regulate RiBi (*Jarzebowski et al., 2018*). In addition, phenotypes of RiBi defects in *Drosophila* GSCs appear to be controversial; attenuation of Pol I activity and the U3 snoRNP complex member *wiched* (*wcd*) mutant show premature differentiation (*Fichelson et al., 2009*; *Zhang et al., 2014*), while knockdown of ribosome assembly and loss of DExD/H-box proteins (Aramis, Athos, and Porthos) that govern RiBi result in a loss of differentiation (*Martin et al., 2022*; *Sanchez et al., 2016*). Therefore, the mechanisms by which RiBi is coupled to low translational activity and proper stem cell differentiation are still obscure.

By using an ENU mutagenesis screen to identify gonadogenesis defects in zebrafish (*Saito et al., 2011*), we isolated a unique mutant, *minamoto* (*moto*), in which germ cells arrest at the early stage of spermatogonia (*Kawasaki et al., 2016*). By whole-genome sequencing (WGS), we found that the *moto* phenotype is tightly linked to a mutation within a gene (ENSDARG00000090664) (*Bowen et al., 2012*), which encodes the coiled-coil domain-containing protein Meioc. Database analysis shows orthologs of MEIOC in vertebrates and most invertebrates, but not in *Drosophila*. The mouse MEIOC functions to maintain an extended meiotic prophase I with its binding partner YTHDC2 (YTH-domain containing 2), which is a 3'–5' RNA helicase containing an RNA helicase motif and YT521-B homology (YTH) RNA-binding domain (*Abby et al., 2016*; *Soh et al., 2017*). On the other hand, a *Drosophila* complex of *bam* (*bag of marbles*), a divergent homolog of *Meioc* sharing conserved amino acid sequences within the coiled-coil domain, and *bgcn* (*benign gonial cell neoplasm*), a paralog of *Ythdc2* (*Jain et al., 2018*), plays a pivotal role in promoting stem cell differentiation (*Perinthottathil and Kim, 2011*). Although zebrafish *meioc* (*moto*) is orthologous to the mice gene, it appears to be functionally similar to *Drosophila bam*, making it interesting to see how it controls the development of germ cells.

The present study revealed that zebrafish had spermatogonial stem cells (SSCs) with low rRNA transcription, a characteristic that differed from many other stem cells, and that Meioc was required for upregulation of rRNA transcription and SSC differentiation. Independently of Ythdc2, Meioc regulated intracellular localization of the Argonaute/Piwi family protein Piwil1 (Piwi-like protein 1) that interacted with transcriptional silencing proteins, Setdb1 (Eggless) and HP1α (Cbx5). Our results suggested that low rRNA transcription maintained the undifferentiated state of zebrafish SSCs, providing a new insight on the relationship between the stem cell system and the RiBi control.

## Results

### Failure to differentiate spermatogonia in the zebrafish *meioc* mutant

In fish testes, germ cells are surrounded by Sertoli cells within a basement membrane compartment and develop synchronously in cysts. Thus, the developmental stage of spermatogonia can be determined by the number of cells within the cyst (*Figure 1A*). Wild-type spermatogonial cells undergo nine rounds of cell division before entering meiosis (*Leal et al., 2009*); however, *moto^{t31533}* mutant testes had only up to three rounds of division, containing 2–8 spermatogonia per cyst (*Figure 1B*). Mutant spermatogonia were positive for the undifferentiated spermatogonia marker, Plzf, which is expressed in single- to 8-cell (1- to 8-cell) cyst spermatogonia in the wild-type (*Figure 1B*; *Ozaki et al., 2011*), but negative for meiotic prophase I markers, Sycp1 and Sycp3 (*Saito et al., 2014*; *Saito et al., 2011*). Histone H3 phospho-S10 immunostaining indicated that mutant spermatogonia proliferated, but apoptotic cells were increased in the mutant spermatogonia (*Figure 1—figure supplement 1A*). In zebrafish, mutants disrupting early stages of oocyte development become male (*Saito et al., 2011*; *Houwing et al., 2007*; *Kamminga et al., 2010*; *Rodríguez-Marí et al., 2010*; *Shive et al., 2010*). Similarly, no females and no oocytes developed in *moto* mutants (*Table 1* and *Figure 1—figure supplement 1B*), as completion of the pachytene stage of meiosis I is a prerequisite for follicle formation (*Elkouby and Mullins, 2017*). These results suggested that the *moto^{t31533}* mutant germ cells remained in an undifferentiated state.

By WGS and genetic analysis, we found that the *moto^{t31533}* mutant phenotype was tightly linked to a mutation within the ortholog of mouse *meioc* gene (*Figure 1—figure supplement 1C*; *Bowen et al.,*

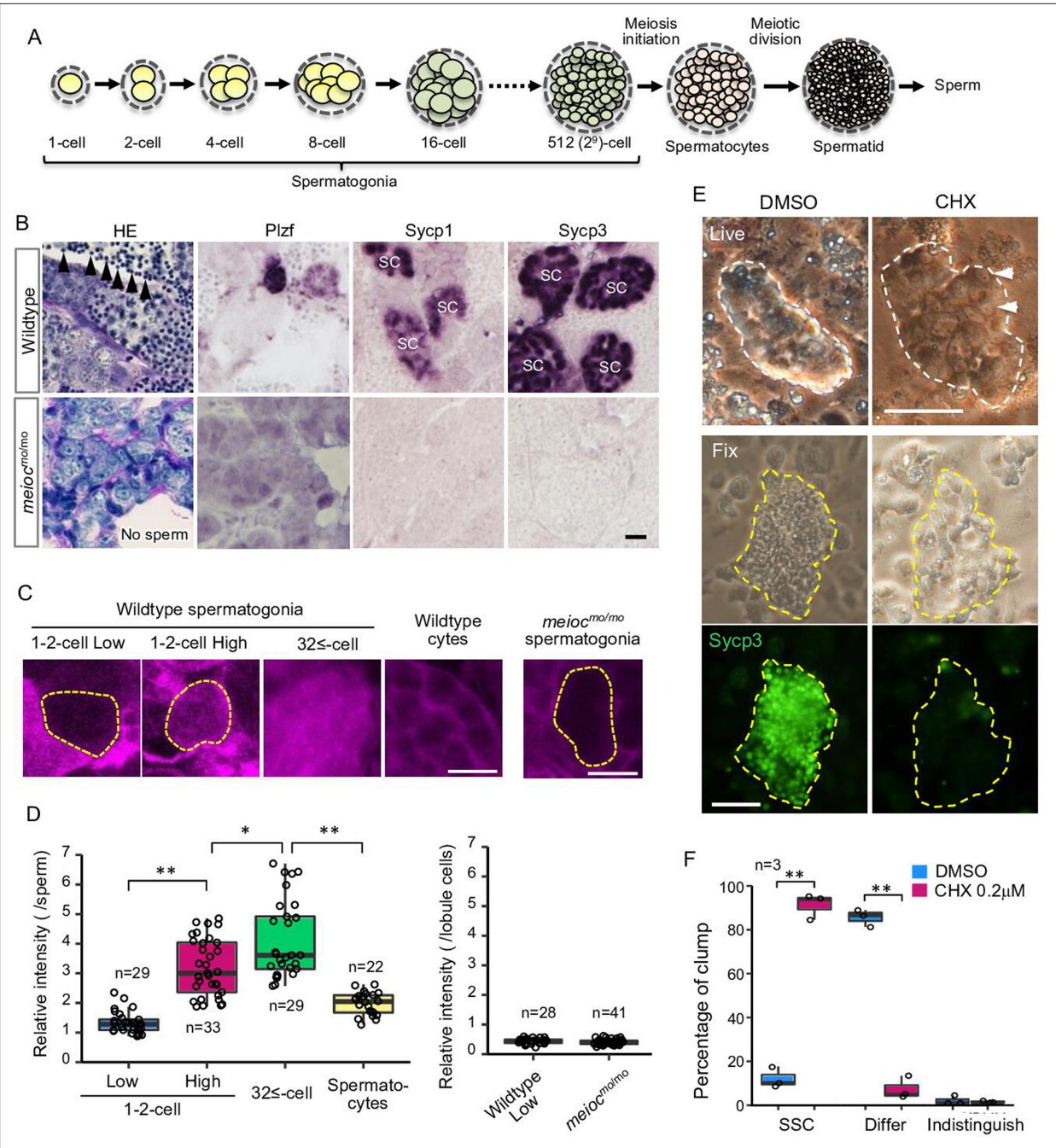

**Figure 1.** Low translational activity of *meioc* mutant spermatogonia. (**A**) Schema of the development of spermatogonial cysts surrounded by Sertoli cells and progression of spermatogenesis in zebrafish. (**B**) Histology (HE) and immunostaining against Plzf and spermatocyte markers (Sycp1, 3) in the wild-type and the *moto*[-/-] testes. Arrowheads: sperm. SC: spermatocytes. Scale bar: 10 μm. (**C**, **D**) OP-Puro fluorescence analysis (**C**) and quantification of the signal intensities (**D**) in wild-type and *meioc*[mo/mo] spermatogenic cells. Dotted lines: 1- to 2-cell spermatogonia. Scale bars: 10 μm. (**E**–**F**) Effect of cycloheximide (CHX, 0.2 μM) on differentiation of spermatogonial stem cells (SSCs) in culture. Dotted lines: germ cell clumps. Sycp3: immunostaining of Sycp3. Arrowheads: examples of a cell with a large nucleolus. Scale bar: 50 μm. The graph (**F**) presents the percentage of clumps of SSCs and differentiated cells (Differ) shown in **E**. Indistinguish: not determined whether dominant cell type was stem or differentiated. Data are represented as mean ± SD. *p<0.05, **p<0.01.

The online version of this article includes the following source data and figure supplement(s) for figure 1:

**Figure supplement 1.** Phenotypes of testes and early gonads of the *moto*[-/-] mutant.

**Figure 1-figure supplement 1-source data 1** .PDF file containing original western blots for *Figure 1—figure supplement 1E*, indicating the relevant bands and treatments.

*Figure 1 continued on next page*

*Figure 1 continued*
**Figure supplement 2.** Expression patterns of Meioc in wild-type germ cells.
**Figure supplement 3.** Effect of cycloheximide on spermatogonia.
**Figure 1-figure supplement 3-source data 1** .PDF file containing original western blots for *Figure 1—figure supplement 3C*, indicating the relevant bands and treatments.

*2012*). Fish harboring the *moto^t31533* mutation inherited the *moto* phenotypes described above for more than 10 generations. Furthermore, fish homozygous for another allele, *moto^sa13122*, exhibited the same gonadal phenotype as *moto^t31533* homozygotes, and *moto^sa13122* failed to complement *moto^t31533*, confirming that these mutations disrupt the same gene (hereafter denoted *meioc^mo*) (*Figure 1—figure supplement 1D*). Both mutations encode a truncated protein that lacks the conserved coiled-coil domain (PF15189). We searched for paralogues of *meioc* gene using PF15189 domain (*Figure 1—figure supplement 1C*) but were not able to find another gene in the zebrafish reference genome assembly GRCz11 (version 111.11). We then generated an antibody against the N-terminus of Meioc (*Figure 1—figure supplement 1E and F*) and could not detect the expression of truncated Meioc in the mutants (*Figure 1—figure supplement 1G*).

Zebrafish *meioc* RNA and protein signals were observed in a portion of 1- to 2-cell cyst germ cells in juvenile gonads at 25 days post-fertilization (dpf), adult ovaries, and testes (*Figure 1—figure supplement 2A*). The cells exhibited a large nucleolus, characteristics of presumed GSCs in zebrafish (*Kawasaki et al., 2016*) and stem-type self-renewing type I germ cells in medaka (*Nishimura et al., 2015*). Both clear RNA and protein signals were detected in premeiotic germ cell clusters, and the protein was detected as granular structures in the cytoplasm with increasing sizes among cells until meiotic leptotene-zygotene stage (*Figure 1—figure supplement 2A–C*).

## Upregulation of translation is required for the differentiation of spermatogonia

Since the global translation activity is generally inhibited in stem cells, we analyzed global translational activities of zebrafish spermatogonia using the *O*-propargyl-puromycin (OP-Puro) assay (*Liu et al., 2012*). In wild-type, we observed two populations, with low levels and high levels of de novo protein synthesis in 1- to 2-cell cyst spermatogonia. Whereas almost all large cysts with more than 32 cells (32≤-cell) of differentiated spermatogonia had high levels of de novo protein synthesis (*Figure 1C and D*). Almost all *meioc^mo/mo* spermatogonia showed similarly low levels as that of 1- to 2-cell cysts (*Figure 1C and D*). These results suggested that some spermatogonia of 1- to 2-cell cysts had low translation activity in wild-type and that *meioc^mo/mo* spermatogonia were arrested in the state of low translation activity.

In order to know if translational upregulation is required for differentiation of spermatogonia, we examined the effect of cycloheximide on the development of spermatogonia in vitro. Cycloheximide decreased OP-Puro fluorescence intensities in differentiating spermatogonia in a dose-dependent manner in testis organ culture (*Figure 1—figure supplement 3A*). At 1.0 μM cycloheximide that reduced OP-Puro fluorescence to approximately 60%, BrdU incorporation decreased in 32≤-cell cysts of spermatogonia, while it was not affected in 1- to 4-cell spermatogonia (*Figure 1—figure supplement 3B and D*). We then examined the effect on differentiation of SSCs using a culture system, in which differentiation can be induced on the Sertoli cell line, ZtA6-12 (*Kawasaki et al., 2016*). After propagation of SSCs for 1 month, we transferred the cells onto ZtA6-12 and treated them with a lower concentration of cycloheximide at 0.2 μM, which reduced OP-Puro fluorescence to approximately 70%. The treatment maintained SSCs with a large nucleolus and not expressing Sycp3, whereas the control

**Table 1.** Number of wild-type, heterozygous, and *meioc* mutant males and females.

|  | Female | Male |
|---|---|---|
| *meioc^mo/mo* | 0 | 24 |
| *meioc^+/mo* | 23 | 14 |
| *meioc^+/+* | 13 | 13 |

without cycloheximide showed expression of Sycp3 (*Figure 1E and F*). Bmp2 production of ZtA6-12 cells was not affected at 0.2 μM cycloheximide (*Figure 1—figure supplement 3C*), suggesting that the effect on Sertoli cell function was minimized. These results suggested that a certain level of translational activity was required for the differentiation of zebrafish SSCs.

## *meioc* mutants do not upregulate rRNAs in 1- to 2-cell cyst spermatogonia

To estimate the state of RiBi, we examined the expression patterns of rRNAs and the ribosomal protein Rpl15 in zebrafish spermatogonia development. Interestingly, the signal intensities of 5.8S, 18S, 28S rRNAs, and Rpl15 were low in a portion of wild-type 1- to 2-cell cyst spermatogonia (*Figure 2A*), suggesting the presence of two populations also in the rRNA concentrations (low and high) in the 1- to 2-cell stage. We did not observe a low intensity of 5S rRNA. Those rRNA signal intensities increased in almost all 32≤-cell cysts and declined in spermatocytes. To distinguish between cytoplasmic and nucleolar signals, we performed fluorescence in situ hybridization of 28S rRNA. We found 1- to 2-cell cysts with low cytoplasmic signals also had low nucleolar signals (*Figure 2B–D*). Although cytoplasmic 28S rRNA signals increased to 32≤-cell cysts, the highest nucleolar signals were detected in portions of 1- to 2-cell cysts and declined at 32≤-cell cysts. These results suggest that 1- to 2-cell cysts contain populations with low and high rRNA transcriptional activity.

We next tested how rRNA levels were affected in *meioc^{mo/mo}* and could not find cells with high signals of rRNAs and Rpl15 in *meioc^{mo/mo}* spermatogonia (*Figure 2A*). We confirmed these data by performing RT-qPCR from isolated spermatogonia using the *sox17::egfp* marker. We have confirmed that *sox17::egfp* spermatogonia undergo self-renewal and differentiation into spermatozoa (*Kawasaki et al., 2016*), and the signal was predominantly expressed in 1- to 2-cell cysts (*Figure 2—figure supplement 1*). Reductions in 5.8S, 18S, and 28S rRNAs in *meioc^{mo/mo}* were observed (*Figure 2E*). Furthermore, the expression of the homolog of *Drosophila* non-LTR retrotransposable element R2 that transposes exclusively into 28S rDNA (*Kojima and Fujiwara, 2004*) was also decreased in *meioc^{mo/mo}*. We next compared levels of unprocessed 45S pre-rRNA and pre-rRNA intermediates by Northern blot between *meioc^{mo/mo}* and the wild-type. Each was normalized to 7SL RNA. In *meioc^{mo/mo}*, the amount of 45S pre-RNA and the intermediates was reduced; however, the ratio of the *meioc^{mo/mo}* to the wild-type was almost the same for 45S pre-rRNA and the intermediates: 0.69 (average 0.49/0.71, mutant/wild-type) of 45S pre-rRNA and 0.61 (0.72/1.19) of the intermediate with the 5′ external transcribed spacer (ETS) probe, and 0.33 (0.05/0.15) and 0.35 (0.17/0.48) with the internal transcribed spacer 1 (ITS1) probe (*Figure 2F*). These results indicate that 45S pre-rRNA transcripts were reduced in *meioc^{mo/mo}*, yet processing into pre-RNA intermediates was not affected.

## Meioc is required for demethylation of CpG in the IGS of 45S rDNA

Features of silenced rDNA loci, DNA methylation, methylation of histone H3K9, and association of HP1α are known (*Grummt, 2007*). In eukaryotes, the transcription of each rDNA locus is regulated by DNA methylation and histone modifications in promoter and enhancer regions in the intergenic spacer (IGS) region (*Santoro, 2005*). Therefore, we examined DNA methylation state in the region. Since the regulatory sequence of rDNA was not identified in teleosts, we identified tandem repeats of 90, 127, and 318 bp in the IGS region of 45S-S rDNA (*Locati et al., 2017*; *Figure 2—figure supplement 2*), similar to carp (*Vera et al., 2003*). Unmethylated CpG in the tandem repeats was frequently found in isolated wild-type *sox17::egfp* spermatogonia, while it was rare in *meioc^{mo/mo}* (*Figure 2G*). These results suggested that Meioc is needed for demethylation of CpG in the tandem repeats of 45S rDNA IGS region.

## *meioc* and *ythdc2* mutants exhibit different phenotypes in the early stage of spermatogonia

The YTHDC2 is a binding partner of the MEIOC in mammals (*Abby et al., 2016*; *Soh et al., 2017*). Zebrafish spermatogenic cells expressed Ythdc2 (*Figure 3—figure supplement 1A and B*), and pull-down analysis showed zebrafish Meioc interacted with Ythdc2 (*Figure 3—figure supplement 1C*). It has been recently reported that *ythdc2* KO zebrafish lack germ cells (*Li et al., 2022*), but staining or expression analysis was not done to confirm that germ cells were completely absent. We separately generated *ythdc2* KO zebrafish by the CRISPR-Cas9 system (*Figure 3—figure supplement 1A*), and

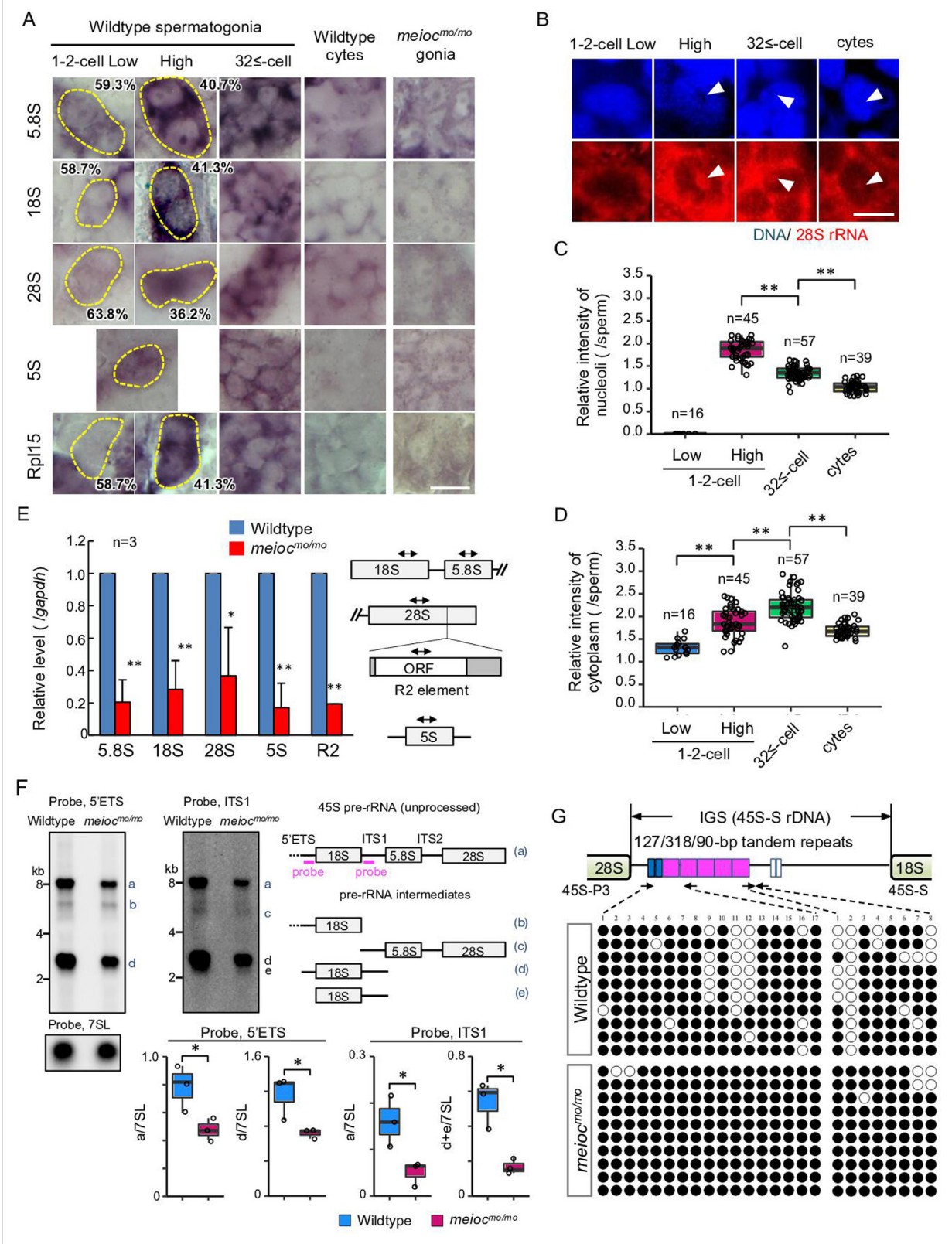

**Figure 2.** Defect on upregulation of rRNA transcription in *meioc^mo/mo^* spermatogonia. (**A**) In situ hybridization of 5S, 5.8S, 18S, and 28S rRNA and immunohistochemistry with anti-Rpl15 antibody in spermatogonia (gonia) and spermatocytes (cyte) in wild-type and *meioc^mo/mo^*. Yellow dotted lines indicate 1- to 2-cell spermatogonia. Percentages represent the frequency of low and high 1- to 2-cell spermatogonia. (**B–D**) Fluorescent in situ hybridization of 28S rRNA in wild-type. The graphs present quantification of signal intensities of 28S rRNA in nucleoli (**C**) and in cytoplasm (**D**) in

*Figure 2 continued on next page*

*Figure 2 continued*

spermatogenic cells. Arrowheads: nucleoli; cytes: spermatocytes. (**E**) qRT-PCR analysis of rRNAs and R2 between wild-type and *meioc^mo/mo^* purified *sox17::egfp* positive spermatogonia. Two-way arrows in the schema indicate the position of primers on the rRNA and R2 element. (**F**) Northern blot analysis of pre-rRNA processing in wild-type and *meioc^mo/mo^* testes using probes for 5' external transcribed spacer (ETS) and internal transcribed spacer 1 (ITS1). Right panel: schema of 45S pre-rRNA and pre-rRNA processing intermediates in zebrafish (*Tao et al., 2017*). Left panel: Northern blot analysis of pre-rRNA processing in wild-type and *meioc^mo/mo^* testes using 5'ETS and ITS1 probes. A probe for the 7SL RNA was used as a loading control. Graphs summarize relative signal intensity of 45S pre-rRNA and intermediates normalized to 7SL in three wild-type and *meioc^mo/mo^* testes. (**G**) Bisulfite-sequencing analysis of the tandem repeat region in the intergenic spacer (IGS) region of the 45S-S rDNA locus in purified undifferentiated spermatogonia of wild-type and *meioc^mo/mo^*. Arrows: position of bisulfite primers in the tandem repeat elements (blue, magenta, and white boxes); black dots: methylated CpG sites; white dots: unmethylated sites. $*p<0.05$, $**p<0.01$. Scale bars: 10 µm.

The online version of this article includes the following source data and figure supplement(s) for figure 2:

**Source data 1.** PDF file containing original Northern blots for *Figure 2F*, indicating the relevant bands and treatments.

**Figure supplement 1.** Expression pattern of EGFP in the *sox17::egfp* transgenic testis.

**Figure supplement 2.** Schema of the intergenic spacer (IGS) (chromosome 5: 826807–831755 reverse strand) and the 127, 318, and 90 bp tandem repeat sequences (**A**) and loci of the tandem repeats (**B**).

observed that it had up to 8-cell cyst spermatogonia (*Figure 3A*). We observed that the number of 4- to 8-cell cysts was clearly different between *meioc^mo/mo^* and *ythdc2^-/-^*; the ratio of 4- to 8-cell cysts/1- to 2-cell cysts in the *ythdc2^-/-^* was almost the same as that in the wild-type, while that in *meioc^mo/mo^* significantly decreased (*Figure 3B–D*). Furthermore, *ythdc2^-/-^* spermatogonia contained both low and high levels of cells with 5.8S, 18S, 28S rRNAs, and Rpl15, similar to the wild-type (*Figure 3E*). The intensity of the 28S rRNA signals in *ythdc2^-/-^* was almost the same as that in the wild-type (*Figure 3— figure supplement 1D*). These results suggested that Meioc functioned independently of Ythdc2 on the differentiation of 1- to 2-cell cysts into 4- to 8-cell cysts.

## Meioc interacts with Piwil1 and affects its intracellular localization

To explore partners of Meioc on regulation of rRNA transcription, we performed LC/MS/MS for the Meioc-immunoprecipitate (IP) of SSC-enriched hyperplastic testes. The hyperplastic testes, which are occasionally found in adult wild-type zebrafish, contain cells at all stages of spermatogenesis. Hyperplasia-derived SSCs self-renewed and differentiated in transplants of aggregates mixed with normal testicular cells (*Kawasaki et al., 2016*). The results showed the enrichment of germ granule (nuage) components, compared with normal testis (*Supplementary files 1 and 2*). By immunostaining, Meioc colocalized with the germ granule components, Tdrd1 (*Huang et al., 2011*), Tdrd6a (*Roovers et al., 2018*), Piwil1 (*Houwing et al., 2007*), Piwil2 (*Houwing et al., 2008*), and Ddx4 (*Houwing et al., 2007*) in wild-type (*Figure 4—figure supplement 1A and B*). In *meioc^mo/mo^*, Piwil1 was strongly detected in the nucleolus (*Figure 4A*, arrowhead), whereas others exhibited the perinuclear localization characteristic of germ granules observed by Ddx4 staining as in the wild-type (*Figure 4A*). In contrast, we were able to detect low levels of Piwil1 in nucleoli of wild-type 1- to 4-cell spermatogonia with overexposed detection after the co-staining of the nucleolar marker fibrillarin (*Figure 4B*). The signals of Piwil1 in Ddx4-positive granules were decreased in *meioc^mo/mo^* spermatogonia, compared with wild-type (*Figure 4C*). We confirmed that Meioc interacted with Piwil1 by co-IP and revealed that Meioc-bound Piwil1 through the coiled-coil domain (*Figure 4D*; *Figure 4—figure supplement 1C*). These results demonstrated that Piwil1 has a property to localize in nucleoli and that Meioc interacts with Piwil1 in perinuclear germ granules. In addition, we did not detect the accumulation of Piwil1 in nucleoli in *ythdc2^-/-^* (*Figure 3—figure supplement 1E*).

Since Piwil1 abnormally accumulated in nucleoli in *meioc^mo/mo^*, we asked whether Piwil1-dependent piRNA generation (*Houwing et al., 2007*) was affected. Total RNA was extracted from *meioc^mo/mo^* and *meioc^+/mo^* testes, and small RNAs (18–35 nt) were sequenced. The abundance of small RNA production in *meioc^mo/mo^* testes was detected at similar levels to those of *meioc^+/mo^* (*Figure 4—figure supplement 1D*). Small RNAs derived from 28S, 18S rRNAs, and R2 transposon apparently decreased in *meioc^mo/mo^*, probably due to the low expression of rRNAs (*Figure 4—figure supplement 1E–G*). Thus, it is unlikely that Meioc critically affects small RNA generation.

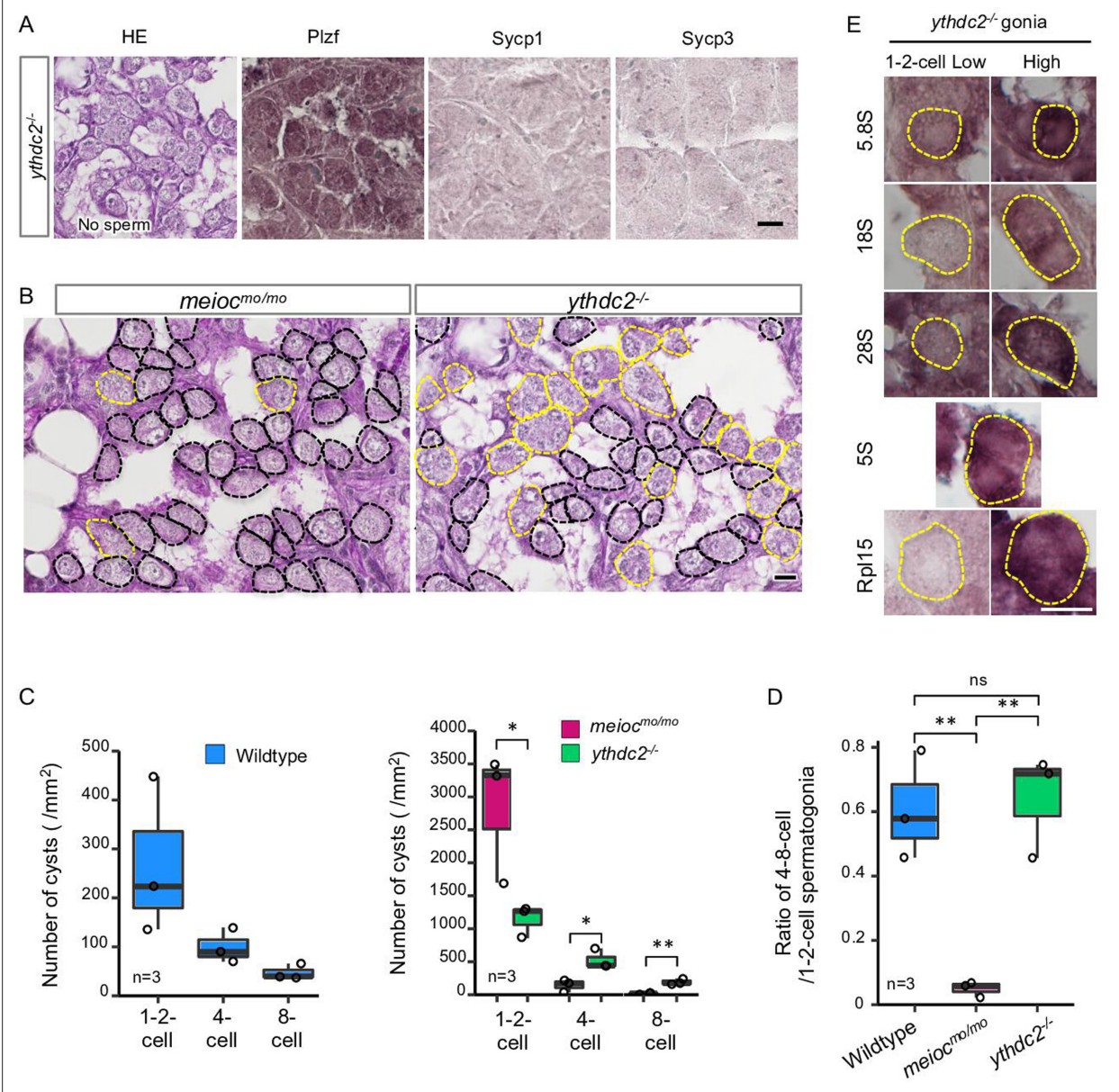

**Figure 3.** *ythdc2* mutant spermatogonia have different defects from *meioc^mo/mo*. (**A**) Histology (HE) and immunostaining against Plzf and spermatocyte markers (Sycp1, 3) in the *ythdc2^-/-* testes. (**B**) Representative image of *meioc^mo/mo* and *ythdc2^-/-* testes sections stained with PAS (periodic acid Schiff) and hematoxylin. Dotted lines: 1- to 2-cell cyst spermatogonia (black) and 4≤-cell cysts (yellow). (**C**) The number of 1- to 2-, 4-, and 8-cell cyst spermatogonia per mm² of sections in wild-type, *meioc^mo/mo*, and *ythdc2^-/-* testes. (**D**) Ratio of the number of 4- to 8-cell cyst spermatogonia to 1- to 2-cell cysts in wild-type, *meioc^mo/mo*, and *ythdc2^-/-* testes. (**E**) In situ hybridization of 5S, 5.8S, 18S, and 28S rRNA and immunohistochemistry with anti-Rpl15 antibody in the *ythdc2^-/-* testes. Yellow dotted lines: 1- to 2-cell spermatogonia. *p<0.05, **p<0.01, ns: not significant. Scale bars: 10 μm.

The online version of this article includes the following source data and figure supplement(s) for figure 3:

**Figure supplement 1.** Mutations of the *ythdc2^-/-*.

**Figure 3-figure supplement 1-source data 1** .PDF file containing original western blots for *Figure 3—figure supplement 1C*, indicating the relevant bands and treatments.

## A reduction of Piwil1 upregulates rRNA transcripts and recovers the *meioc* mutant phenotype

From the above analysis, it is hypothesized that accumulation of Piwil1 in the nucleolus suppresses rRNA transcription in the *meioc^mo/mo* background. Therefore, we examined the rRNA transcript levels and the phenotype of *meioc^mo/mo;piwil^+/-* since the *piwil1^-/-* are viable but deplete germ cells completely

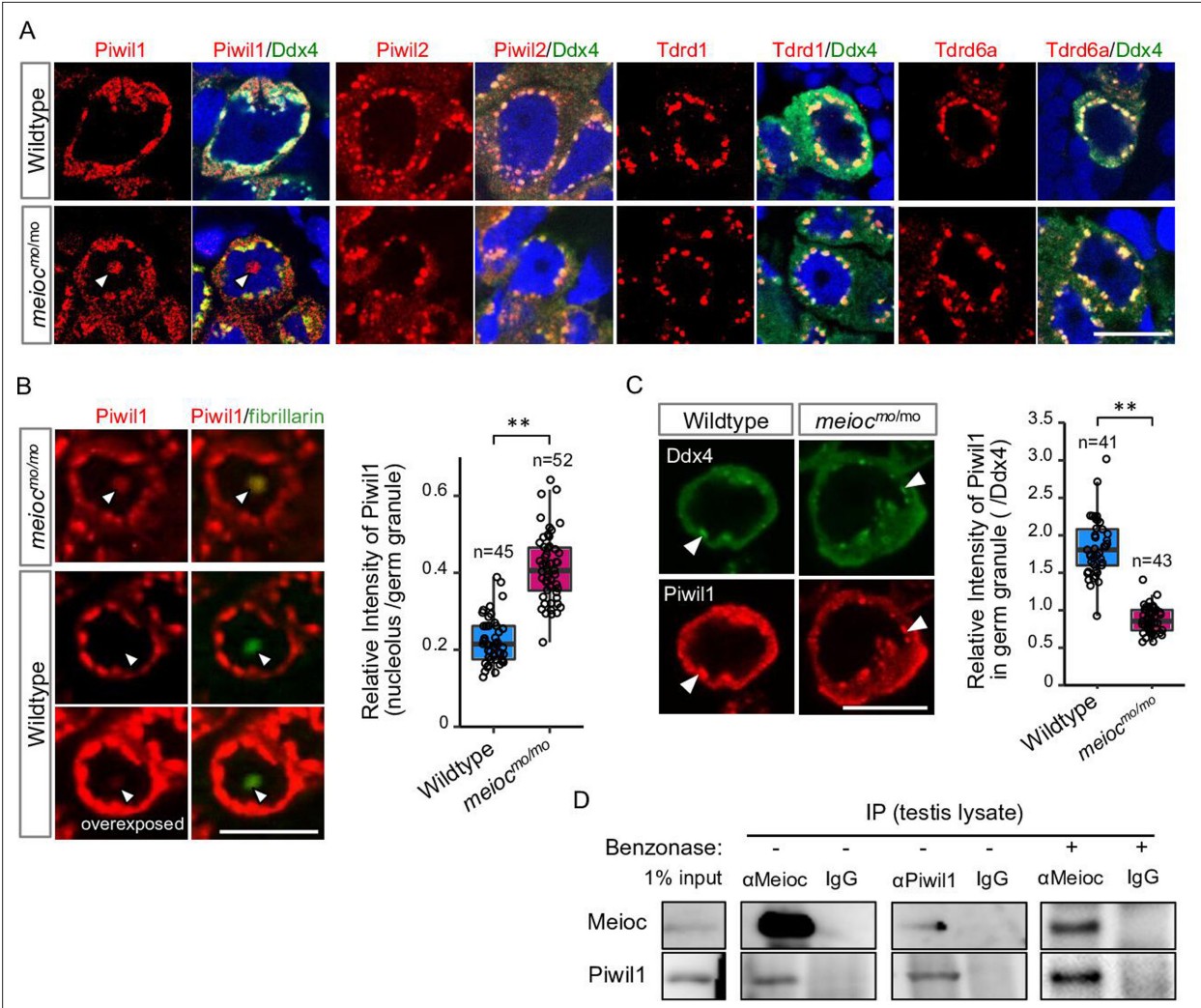

**Figure 4.** Meioc binds with Piwil1 and affects the localization of Piwil1. (**A**) Immunostaining of Ddx4 and Piwil1, Piwil2, Tdrd1, and Tdrd6a in wild-type and *meioc^{mo/mo}* spermatogonia. The arrowhead: the Piwil1 signal in the nucleolus. (**B**) Immunostaining against Piwil1 and fibrillarin (left panels) and quantification of nucleolar Piwil1 (right panel) in wild-type and *meioc^{mo/mo}* spermatogonia. Arrowheads: fibrillarin positive nucleolus. (**C**) Immunostaining of Piwil1 and Ddx4 (left panels) and quantification of Piwil1 in germ granules (right panel) in wild-type and *meioc^{mo/mo}* spermatogonia. Arrowheads: Ddx4 positive germ granules. (**D**) Co-immunoprecipitation of Meioc and Piwil1 using testis lysate. Meioc signals were detected in Piwil1 immunoprecipitate and vice versa. Benzonase: addition of benzonase nuclease. **p<0.01. Scale bars: 10 µm.

The online version of this article includes the following source data and figure supplement(s) for figure 4:

**Source data 1.** PDF file containing original western blots for *Figure 4D*, indicating the relevant bands and treatments.

**Figure supplement 1.** Immunostaining of Meioc with germ granule components and piRNA profiles in the *meioc^{mo/mo}* testis.

**Figure 4-figure supplement 1-source data 1** .PDF file containing original western blots for *Figure 4—figure supplement 1C*, indicating the relevant bands and treatments.

before testis differentiation (*Houwing et al., 2007*). A reduction in Piwil1 was detected in *piwil^{+/-}* testes by western blot analysis and in the spermatogonia nucleoli by fluorescent immunohistochemistry (*Figure 5—figure supplement 1A and B*). The upregulation of 28S rRNA was observed in the cytoplasm and nucleoli of *piwil1^{+/-}* 1- to 2-cell spermatogonia as compared to wild-type and *meioc^{mo/mo};piwil1^{+/-}* spermatogonia as compared to *meioc^{mo/mo}* (*Figure 5A and B*). Consistent with the increased expression of 28S rRNA, we observed the differentiation of *meioc^{mo/mo};piwil1^{+/-}* spermatogonia to the 32-cell stage compared to that of *meioc^{mo/mo}*, in which spermatogonia rarely develop to the 8-cell stage (*Figure 5C and D*). Furthermore, the number of 4- to 8-cell cysts increased in *meioc^{mo/mo};piwil1^{+/-}* (*Figure 5E and F*). The ratio of 4- to 8-cell to 1- to 2-cell cysts in *meioc^{mo/mo};piwil1^{+/-}* increased compared

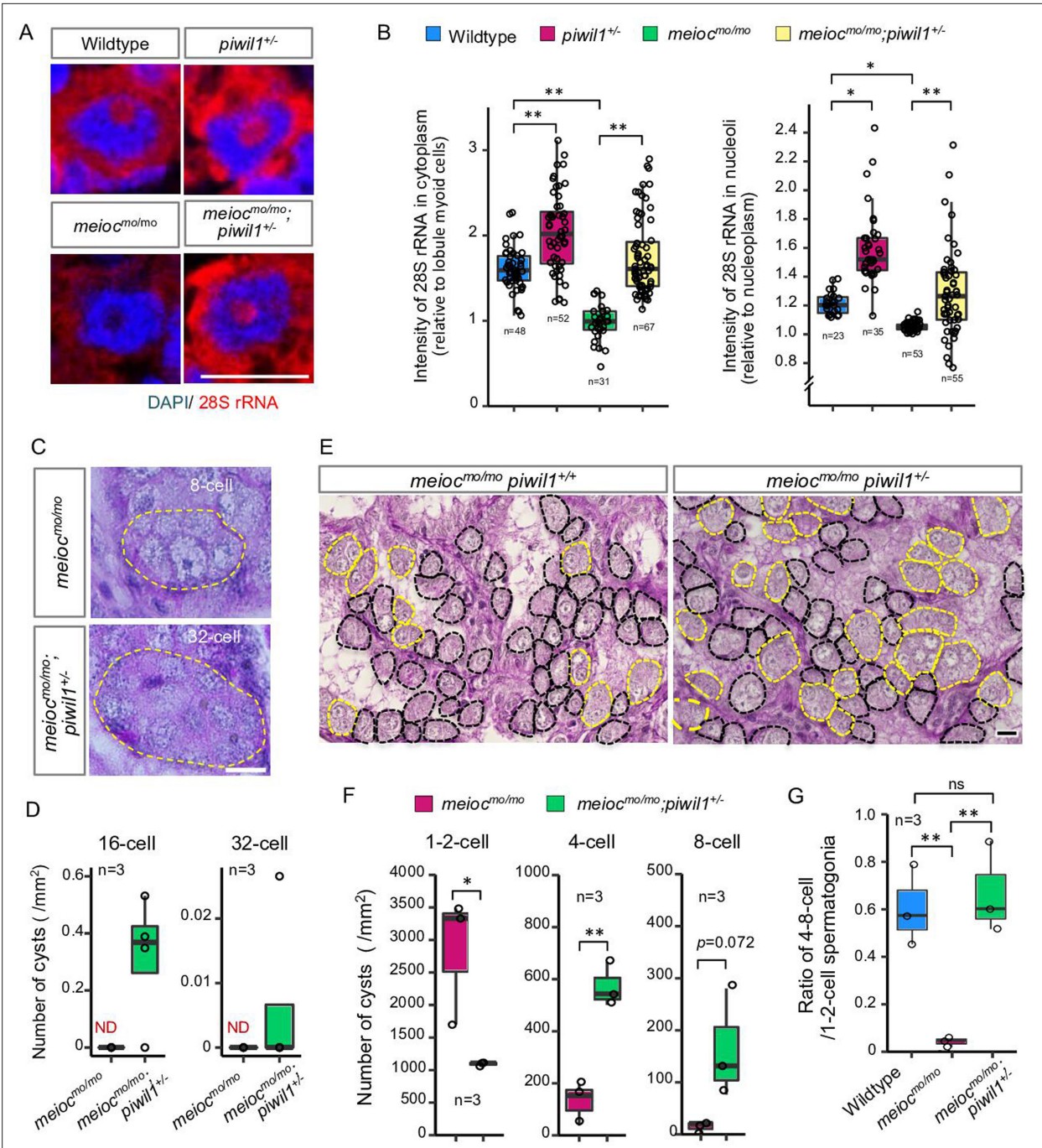

**Figure 5.** Reduction of Piwil1 compensated phenotypes of *meioc^mo/mo^*. (**A, B**) In situ hybridization of 28S rRNA in wild-type and *meioc^mo/mo^;piwil1^+/-^* spermatogonia (1- to 2-cell cysts). Graphs (**B**) show the relative signal intensity in the cytoplasm normalized to the intensity of lobule myoid cells (left) and nucleoli normalized to the intensity of the nucleoplasm (right). (**C, D**) Differentiated spermatogonia in *meioc^mo/mo^* and *meioc^mo/mo^;piwil1^+/-^* testes. Yellow dotted lines: differentiated spermatogonia. Graphs (**D**) show the number of 16-cell and 32-cell cyst spermatogonia per mm² of sections. ND: not detected. (**E–G**) *meioc^mo/mo^* and *meioc^mo/mo^ piwil1^+/-^* testis sections stained with PAS and hematoxylin. Cysts of 1- to 2-cell spermatogonia (black) and 4≤-cell cysts (yellow) are indicated by dotted lines. Graphs show numbers of 1-, 2-, 4-, and 8-cell cysts per mm² in sections of *meioc^mo/mo^* and *meioc^mo/mo^;piwil1^+/-^* testes (**F**), and ratio of the number of 4- to 8-cell cysts to 1- to 2-cell cysts in wild-type, *meioc^mo/mo^* and *meioc^mo/mo^;piwil1^+/-^* (**G**). *p<0.05, **p<0.01, ns: not significant. Scale bars: 10 μm.

The online version of this article includes the following source data and figure supplement(s) for figure 5:

**Figure supplement 1.** Reduction of Piwil1 in *piwil1^+/-^*.

**Figure 5-figure supplement 1-source data 1** .PDF file containing original western blots for *Figure 5—figure supplement 1A*, indicating the relevant bands and treatments.

to that in *meioc^mo/mo* and reached almost the same level as that in wild-type (*Figure 5G*). These results demonstrated that reduction of Piwil1 compensated the suppression of rRNA transcription and the SSC differentiation defect of the *meioc^mo/mo* phenotypes.

## Piwil1 interacts with nascent 45S pre-rRNA in nucleoli of SSCs

To investigate if nucleolar Piwil1 interacts with 45S pre-rRNA, we examined pre-rRNA in the RNA IP (RIP) of Piwil1 by qRT-PCR. Pre-rRNA was associated with Piwil1 compared with the control IgG in the wild-type testis and was more enriched in *meioc^mo/mo* (*Figure 6A*). Furthermore, we examined the effect of actinomycin D (RNA polymerase I [Pol I] and Pol II inhibitor), BMH-21 (Pol I inhibitor) (*Colis et al., 2014*), and α-amanitin (Pol II inhibitor) (*Bensaude, 2011*) on nucleolar localization of Piwil1 in the *meioc^mo/mo* testis organ culture. The nucleolar signals of Piwil1 declined upon treatment with actinomycin D and BMH-21 but not α-amanitin, suggesting that the pre-rRNA transcript was involved in Piwil1 localization to the nucleolus (*Figure 6B*). Together, these results suggested that Piwil1 interacted with nascent pre-rRNA transcripts in nucleoli of SSCs.

## Nucleolar Piwil1 causes accumulation of H3K9me3 and HP1α

*Drosophila* Piwi requires histone methyltransferase Setdb1 to lead to H3K9me3 deposition, HP1α accumulation, and heterochromatin formation for silencing of Pol II-mediated transcription (*Jia et al., 2022*). We examined if Setdb1 and HP1α are localized in nucleoli using anti-Setdb1 (*Figure 6—figure supplement 1A*) and anti-HP1α (*Figure 6—figure supplement 1B and C*) antibodies. Setdb1 was detected in nucleoli in spermatogonia of 1- to 2-cell cysts in both wild-type and *meioc^mo/mo*, and co-IP using *meioc^mo/mo* testis lysate showed interaction between Piwil1 and Setdb1 (*Figure 6C and D*). A higher signal intensity of Setdb1 nucleolar signal was detected in *meioc^mo/mo* spermatogonia (*Figure 6E*). To test whether the amount of nucleolar Piwil1 correlates with the silencing state of rDNA loci, we conducted chromatin IP of histone H3K9me3 and Piwil1. Higher enrichment of H3K9me3 and Piwil1 with rDNA was detected in *meioc^mo/mo* testes, which have increased levels of nucleolar Piwil1, than those in the wild-type (*Figure 6F and G*). In addition, decreased enrichment was detected in *piwil1^+/-* testes, which have a low amount of nucleolar Piwil1 (*Figure 6F and G*). Furthermore, HP1α was detected in nucleoli in spermatogonia of 1- to 2-cell cysts in wild-type and *meioc^mo/mo*, and co-IP using testis lysate and pull-down assay showed an interaction between Piwil1 and HP1α (*Figure 6H and I*; *Figure 6—figure supplement 1D*). HP1α was predominantly localized in nucleoli in wild-type with higher nucleolar levels in *meioc^mo/mo* (*Figure 6J*). These results indicated that Piwil1 potentially interacted with Setdb1 in nucleoli and led to an increased H3K9me3 modification and HP1α accumulation in rDNA loci in zebrafish spermatogonia.

## Meioc expression is correlated with the upregulation of 28S rRNA

To further investigate the relationship with Meioc and upregulation of rRNA in SSCs, we compared the expression patterns of Meioc, 28S rRNA, and Piwil1 in isolated *sox17::egfp* spermatogonia. Meioc expression ranged from barely detectable to more than 51 (51≤) granular dots (*Figure 7A*). Nuclear localization of Meioc was observed only in some *sox17::egfp* spermatogonia with 51≤Meioc granules, while in other spermatogonia and spermatocytes Meioc was detected exclusively in the cytoplasm (*Figure 1—figure supplement 2C*, *Figure 7—figure supplement 1A*). Therefore, we analyzed *sox17::egfp* spermatogonia with 51≤Meioc granules by separating those in the nucleus from those in the cytoplasm. The 28S rRNA intensity increased in cells with 11–50 Meioc granules and more (*Figure 7A*). On the other hand, Piwil1 expression levels were apparently the same across most cells, except cells with 51≤nuclear Meioc granules (*Figure 7—figure supplement 1B*). Since nucleoli were difficult to observe in the isolated cells, probably due to the isolation procedure, which takes several hours, it was difficult to verify whether Piwil1 was present in the nucleolus when Meioc expression was low. Therefore, we also examined the expression of Piwil1 in 1- to 2-cell spermatogonia in wild-type testis sections. Piwil1 was detected at almost the same levels in all cells similar to the isolated cells (*Figure 7B*). The large nucleolus, characteristics of presumed GSCs as described above, was observed in the cells with 11–50 Meioc granules and more, and nucleolar localization of Piwil1 was observed in the cells with 11–50 Meioc granules, but not in the cells with 51≤Meioc granules. These results support the idea that Meioc inhibits localization of Piwil1 in nucleoli and that Meioc functioned on upregulation of rRNA transcripts by preventing localization of Piwil1 in nucleoli.

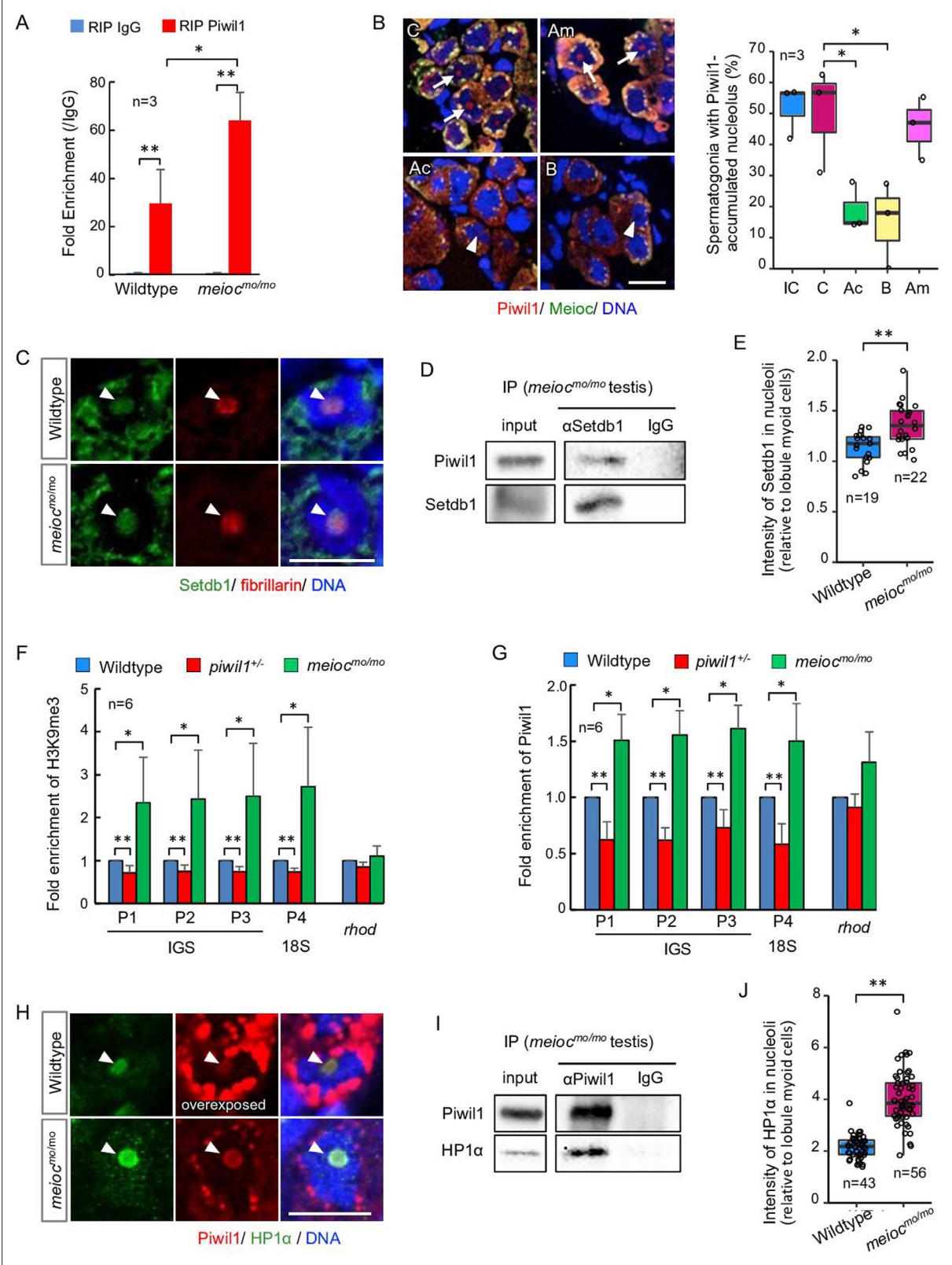

**Figure 6.** Nucleolar Piwil1 interacted with Setdb1 and caused silenced epigenetic state of rDNA loci. (**A**) Fold enrichment of pre-rRNA (5'ETS-18S rRNA) in Piwil1 immunoprecipitated RNA relative to the control IgG in wild-type and *meioc^mo/mo* testes. (**B**) Immunostaining of Piwil1 (left panels) and the percentage of spermatogonia with detectable nucleolar Piwil1 (right panel) in the *meioc^mo/mo* testes treated with α-amanitin (Am), actinomycin D (Ac), and BMH-21 (B). Arrows: Piwil1 detectable nucleoli; arrowheads: Piwil1 undetectable nucleoli; C: control without inhibitors; IC: initial control.

*Figure 6 continued on next page*

*Figure 6 continued*

(**C**) Immunostaining of Setdb1 and fibrillarin in wild-type and *meioc*<sup>mo/mo</sup> spermatogonia. Arrowheads: nucleoli. (**D**) Co-IP of Piwil1 and Setdb1 using *meioc*<sup>mo/mo</sup> testes lysate. Piwil1 was detected in Setdb1 IP. (**E**) Intensities of Setdb1 in nucleoli in wild-type and *meioc*<sup>mo/mo</sup> spermatogonia. (**F, G**) ChIP-qPCR analysis of H3K9me3 (**F**) and Piwil1 (**G**) levels in 45S-rDNA region in wild-type, *piwil1*<sup>+/-</sup>, and *meioc*<sup>mo/mo</sup> testes. The position of primers was indicated in *Figure 2—figure supplement 2*. Mean ± SD are indicated. (**H**) Immunostaining of HP1α and Piwil1 in wild-type and *meioc*<sup>mo/mo</sup> spermatogonia. Arrowheads: nucleolus. (**I**) Co-IP of Piwil1 and HP1α using *meioc*<sup>mo/mo</sup> testis lysate. HP1α was detected in Piwil1 IP. (**J**) Intensities of HP1α in nucleoli in wild-type and *meioc*<sup>mo/mo</sup> spermatogonia. *p<0.05, **p<0.01. Scale bars: 10 μm.

The online version of this article includes the following source data and figure supplement(s) for figure 6:

**Source data 1.** PDF file containing original western blots for *Figure 6D*, indicating the relevant bands and treatments.

**Source data 2.** PDF file containing original western blots for *Figure 6I*, indicating the relevant bands and treatments.

**Figure supplement 1.** Interaction of Piwil1 with Setdb1 and HP1α.

**Figure 6-figure supplement 1-source data 1** .PDF file containing original western blots for *Figure 6—figure supplement 1A*, indicating the relevant bands and treatments.

**Figure 6-figure supplement 1-source data 2** .PDF file containing original western blots for *Figure 6—figure supplement 1B*, indicating the relevant bands and treatments.

**Figure 6-figure supplement 1-source data 3** .PDF file containing original western blots for *Figure 6—figure supplement 1C*, indicating the relevant bands and treatments.

**Figure 6-figure supplement 1-source data 4** .PDF file containing original western blots for *Figure 6—figure supplement 1D*, indicating the relevant bands and treatments.

## Discussion

This study showed that zebrafish had SSCs with low rRNA transcription activity, and that increased activity correlated with differentiation. The state of low rRNA transcription activity is not similar to high RiBi of ES cells or *Drosophila* GSCs, but is closer to the low expression of ribosomal gene of quiescent NSCs (*Llorens-Bobadilla et al., 2015*). Since zebrafish SSCs have long-term BrdU-retaining cells (*Nóbrega et al., 2010*), low RiBi may be one of the states in quiescent cells, although quiescent HSCs have significant RiBi (*Jarzebowski et al., 2018*). Zebrafish 1- to 2-cell cyst spermatogonia had low and high nucleolar 28S rRNA subpopulations, and the high rRNA cells were not found in *meioc* mutants. Therefore, high rRNA is considered to be an activated state for differentiation. The progenitor 32≤-cell cysts had the highest translational activity and lower rRNA transcriptional activity than high rRNA 1- to 2-cell cysts.

Our results also revealed that Meioc was required for the reduction of H3K9me3 levels and demethylation of CpG at rDNA loci. As shown in *Figure 7C*, Meioc prevented nucleolar localization of Piwil1. Piwil1 interacted with Setdb1, and both proteins could be found in the nucleolus. Furthermore, high levels of H3K9me3 and HP1α correlated with higher levels of nucleolar Piwil1. This is similar to *Drosophila* Piwi in piRNA-dependent gene silencing for Pol II transcribed genes (*Jia et al., 2022*). These data support the model that nucleolar Piwil1 interacts with Setdb1 in the nucleolus to maintain low levels of rRNA transcription via H3K9me3 and recruiting HP1α while Meioc promotes differentiation by blocking Piwil1 nucleolar localization and allowing rRNA transcription. The epigenetic state of tandem repeats of the zebrafish rDNA IGS region was similar to human silenced rDNA with CpG hypermethylation, levels of H3K9me3, and HP1α association (*Grummt, 2007*). The tandem repeats have been reported as enhancers of rRNA in *Xenopus* (*Pikaard and Reeder, 1988*), and the methylation status of the region is accompanied by rRNA synthesis (*Bird et al., 1981*). Since Setdb1 associates with Tip5, which is known to interact with DNA methyltransferase as a component of nucleolar remodeling complex (NoRC)(*Yuan et al., 2007*; *Bersaglieri and Santoro, 2019*), the Setdb1 that is interacted with nucleolar Piwil1 in this study is presumed to lead to CpG methylation in rDNA IGS region.

The phenotypes of the zebrafish *meioc* mutants and *ythdc2* mutants are different from those of mouse *Meioc* KO and *Ythdc2* KO, which arrest at meiotic prophase I or immature follicles (*Abby et al., 2016*; *Soh et al., 2017*; *Wojtas et al., 2017*). In addition to different timing of phenotype expression between mice and zebrafish, the amino acid sequence of Meioc between mice and zebrafish is not well conserved (28.5%), while that of Ythdc2 is well conserved (65.5%). It is likely that zebrafish Meioc have different binding partners and regulate different processes of germ cell development from mouse MEIOC, causing the distinct phenotypes between these mutants. Many unique partners

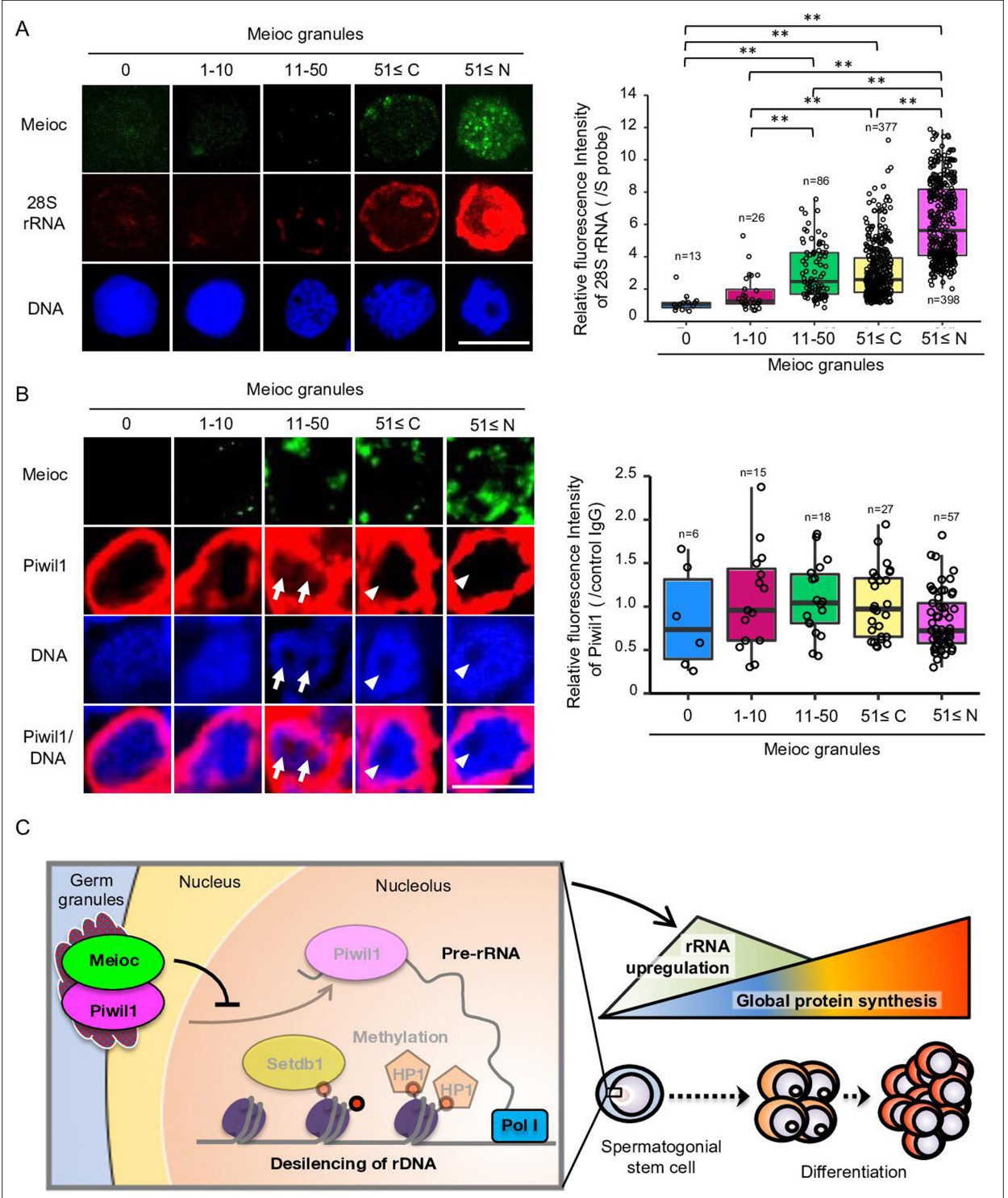

**Figure 7.** Meioc was required for upregulation of 28S rRNA. (**A**) Expression pattern of 28S rRNA in isolated *sox17::egfp* spermatogonia, based on the amount of Meioc granules and the localization. Right panels are intensities of 28S rRNA for each class of the purified *sox17::egfp* spermatogonia. 51≤C and N: 51≤cytoplasmic and nuclear Meioc granules, respectively. *p<0.05, **p<0.01, n: number of analyzed spermatogonia. Scale bars: 10 μm. (**B**) Expression patterns of Piwil1 in five classes of Meioc-expressing spermatogonia in testis sections. The graph shows intensities of Piwil1 in wild-type testes sections for each class of the spermatogonia. The characteristic large nucleoli of spermatogonial stem cells (SSCs) were identified based on nucleolar voids observed in DAPI staining. Arrows: Piwil1 detectable nucleoli; arrowheads: Piwil1 undetectable nucleoli. 51≤C and N: 51≤cytoplasmic and nuclear Meioc granules, respectively. **p<0.01. Scale bars: 10 μm. (**C**) Graphical abstract. Meioc prevents the nucleolar localization of Piwil1 and its associated Setdb1 and HP1α to upregulate rRNA transcripts that are required for zebrafish SSCs to differentiate.

*Figure 7 continued on next page*

*Figure 7 continued*

The online version of this article includes the following figure supplement(s) for figure 7:

**Figure supplement 1.** Intracellular localization of Meioc in spermatogenic cells and Piwil1 expression levels in isolated *sox17::egfp* spermatogonia.

of Meioc, such as RNA-binding proteins, myosin proteins, Tudor domain proteins, heat shock proteins, and Cnot1, were found in the present study.

In this study, we revealed the function of Piwil1 in regulating rRNA transcription. In *Drosophila*, nucleolar localization of Piwi and impairment of its localization by inhibition of RNA Pol I in ovarian somatic cells and nurse cells, and accumulation of undifferentiated GSC, delay of GSC proliferation, and upregulation of R1, R2, and rRNA transcripts in *piwi* mutants have been reported (*Fefelova et al., 2017*; *Mikhaleva et al., 2019*; *Mikhaleva et al., 2015*; *Yakushev et al., 2016*). In *Caenorhabditis elegans*, the nuclear Argonaute protein NRDE-3 binds to ribosomal siRNAs (risiRNA) and is translocated from the cytoplasm to the nucleolus, in which the risiRNA/NRDE complex associates with pre-RNAs and reduces the level of pre-rRNAs (*Zhou et al., 2017b*; *Zhou et al., 2017a*; *Zhu et al., 2018*). Therefore, Argonaute proteins appear to have a conserved function on controlling RiBi. This study showed that zebrafish Piwil1 used the analogous machinery for rRNA transcription as Piwi-dependent silencing Pol II transcription in *Drosophila*. Since Meioc was identified as a component of germ granules, the novel function of Piwil1 will open new insights into the interactions of germ granules and nucleoli to regulate RiBi and SSC differentiation.

## Materials and methods
### Zebrafish
The *moto^{t31533}* mutant fish were isolated in the Tubingen line by *N*-ethyl-*N*-nitrosourea mutagenesis screening as described (*Saito et al., 2011*). The *moto^{sa13122}* mutants were isolated by the Zebrafish Mutation Project (*Kettleborough et al., 2013*) and provided by the Zebrafish International Resource Center (ZIRC), Eugene, Oregon. We used the *piwil1^{hu2479}* line (*Houwing et al., 2007*), *India* line, *AB* line, *IM* line (*Shinya and Sakai, 2011*), *vas::egfp* (*Krøvel and Olsen, 2002*), and *sox17::egfp* (*Mizoguchi et al., 2008*) transgenic fish. The use of these animals for experimental purposes was approved by the committee on laboratory animal care and use at the National Institute of Genetics (approval identification numbers 27-12 and 28-13) and the University of Massachusetts Boston Institutional Animal Care and Use Committee (protocol #20120032), and carried out according to the Animal Research Reporting of In Vivo Experiments (ARRIVE) guidelines and to relevant guidelines and regulations.

### Identification of the meioc gene
We used WGS to map the *moto^{t31533}* mutation to an approximately 19 Mb (~14 cM) region on chromosome 3 (*Bowen et al., 2012*), revealing two nonsynonymous changes within this region. One was a missense mutation affecting the *smg1* gene (ENSDARG00000054570), and the other was a nonsense mutation affecting ENSDARG00000090664, orthologue of human C3H17orf104, now called *Meioc* (*Abby et al., 2016*). By further recombination mapping of the *moto^{t31533}* mutation using simple sequence length polymorphisms, the mutation was found in a region of about 2.1 Mb, between markers z22516 and z8680, which contained the C3H17orf104 locus and excluded the *smg1* locus.

### The ythdc2 mutant
The *ythdc2^{-/-}* fish were generated by CRISPR-Cas9 mutagenesis based on protocols (*Chen et al., 2017*; *Hwang et al., 2013*). A single-guide RNA (*Supplementary file 3*) was designed to target exon 5 of *ythdc2* (ENSDART00000166268.2) to delete functional domains predicted by Pfam 35.0 (*Figure 3—figure supplement 1A*; *Mistry et al., 2021*). *ythdc2* sgRNA (100 ng/μl) and 10 pmol/μl Cas9 NLS protein (abm) were co-injected into 1-cell stage *India* embryos. Founders were backcrossed with *India* fish, and the F1 siblings were screened by genotyping. Heterozygous *ythdc2* knockout carrying a −14 bp frameshift mutation in exon 5 (*ythdc2^-*, a −14 bp deletion affecting the codons from G206 that generates 63 amino acid residues from the wrong frame and stop codon after the 269th amino acid) was obtained.

## Meioc and Ythdc2 antibodies

To produce specific antibodies against Meioc and Ythdc2, *meioc* cDNA encoding 356 amino acid residues from N-terminus and *ythdc2* cDNA encoding amino acid residue Arg743 to Leu1381 were cloned into a pQE-30 vector (QIAGEN) and a pET-21a (+) vector (Novagen), respectively. The 6x histidine tag-fused proteins were expressed and purified as described previously (*Ozaki et al., 2011*). Rats and rabbits were immunized with the purified Meioc and Ythdc2 recombinant proteins, respectively. Then, anti-Meioc rat IgG and anti-Ythdc2 rabbit IgG were purified with CNBr-activated sepharose (Cytiva) conjugated with recombinant proteins according to the manufacturer's instructions.

## Western blot analysis

Western blot analyses were performed as described (*Ozaki et al., 2011*) using antibodies (*Supplementary file 4*). Chemiluminescent signals generated with ECL Prime (GE Healthcare) were detected and quantified with Chemidoc XRS Plus (Bio-Rad). For the quantification of Piwil1, eight wild-type testes and nine *piwil1*$^{+/-}$ testes were individually used for the protein extraction and western blotting analysis. The amount of Piwil1, BMP-2, and α-Tubulin was quantified using quantity tools of ImageLab software version 6.0.1 (Bio-Rad), and Piwil1 and BMP-2 signal intensities were normalized with signal intensities of α-Tubulin.

## Histological observation

Testes and juveniles were fixed in 4% PFA in PBS or Bouin's solution (Sigma) for 2 hr at room temperature (RT). Paraffin sections were prepared at a 5 μm thickness and stained with hematoxylin and eosin. For the count of spermatogonial cysts, complete serial sections of three testes each of wild-type, *ythdc2*$^{-/-}$, *meioc*$^{mo/mo}$; *piwil1*$^{+/+}$, and *meioc*$^{mo/mo}$; *piwil1*$^{+/-}$ were stained with PAS (periodic acid Schiff) to stain the Sertoli cells (*Saito et al., 2014*). The number of spermatogonia in cysts was identified by observation of adjacent sections. For wild-type testes, we counted them in randomly selected 10 sections. Undifferentiated spermatogonia (1- to 8-cell spermatogonia) in *ythdc2*$^{-/-}$, *meioc*$^{mo/mo}$; *piwil1*$^{+/+}$, and *meioc*$^{mo/mo}$; *piwil1*$^{+/-}$ testes were counted in randomly selected 10 fields (23547.2 μm$^2$/field) of sections, and estimated average number of each stage of spermatogonia. The area of the sections of wild-type testes and the fields used for the counting was calculated by using ImageJ/Fiji software (*Schindelin et al., 2012*).

## Immunohistochemistry and in situ hybridization in testis sections

Immunohistochemistry was performed with slight modifications as described (*Kawasaki et al., 2016*). Rehydrated sections were antigen retrieved using ImmunoSaver reagent (Nisshin EM) as per the manufacturer's instructions and blocked with EzBlockChemi (Atto) containing 5% BSA (Sigma). Used antibodies and reagents were listed in *Supplementary file 4*. To analyze Ddx4, Piwil1, Piwil2, Tdrd1, and Tdrd6a localization in *meioc*$^{mo/mo}$ spermatogonia, we used anti-Ddx4 IgG labeled with fluorescein using Fluorescein labeling kit – NH2 (Dojindo) – and performed double staining with other antibodies because all antibodies were generated in rabbits.

In situ hybridization of rRNAs was performed with slight modifications as described (*Ozaki et al., 2011*). To synthesize digoxingenin (DIG) labeled cRNA probes, cDNA of rRNAs was amplified from RT-PCR of testicular RNA using primer sets (*Supplementary file 3*) and cloned into the pGEM-T Easy vector (Promega). Twelve loci of precursor 45S rRNA were identified in zebrafish genome (*Locati et al., 2017*), and we designed the specific primer sets that are able to detect rRNAs derived from more than six loci. The DIG-labeled cRNA probes were synthesized using the DIG RNA labeling kit (Roche). The reagents and antibodies were listed in *Supplementary file 4*.

For the fluorescence detection, images were obtained under an FV1000 confocal microscope (Olympus). Overexposed Piwil1 images were acquired under conditions of high detector sensitivity, ignoring halation of Piwil1 signals in cytoplasm. The signal strength was quantified using ImageJ/Fiji software (*Schindelin et al., 2012*). Three testes of wild-type, *meioc*$^{mo/mo}$, *piwil1*$^{+/-}$, and *piwil1*$^{+/-}$ were used for the quantification. The signal intensities were normalized to the intensities of neighboring sperm or the intensities of myoid cells of basement membrane of lobules.

## TdT-mediated nick-end labeling

TdT-mediated nick-end labeling assays were performed using in situ cell death detection kit (AP; Roche, Germany) as described by the manufacturers. The experiments were repeated three times.

## Whole-mount in situ hybridization and immunohistochemistry

Whole-mount in situ hybridization and immunohistochemistry were performed as described (*Nakamura et al., 2006*). A cDNA clone for *meioc* was isolated from RT-PCR of testicular RNA using the primer set (*Supplementary file 3*). The DIG-labeled RNA probes were synthesized using the *meioc* cDNA and DIG RNA-labeling kit (Roche), and the reagents and antibodies were listed in *Supplementary file 4*.

## Protein synthesis assay

Measurement of protein synthesis of germ cells was performed by the Click-iT Plus OPP Alexa Fluor 594 protein synthesis assay kit (Molecular Probes) with slight modifications as described (*Sanchez et al., 2016*). The fragments of the *meioc^{mo/mo}* and wild-type *sox17::egfp* testes were treated with Leibovitz's L-15 medium (Sigma) containing a 1:400 dilution of Click-iT OPP reagent at 28°C for 30 min. Fluorescence images were acquired with a confocal microscope (FV1000, Olympus). Quantification of OP-Puro fluorescence intensity was performed using ImageJ as described (*Sanchez et al., 2016*). The signal intensities were normalized to the intensities of neighboring sperm and the myoid cells of basement membrane of lobules. The experiments were repeated three times.

## Inhibition of ribosome translation in culture

Testis fragments were cultured on the floating membrane in the spermatogonia proliferation medium without growth factors (*Kawasaki et al., 2012*), with cycloheximide at 0.1 µM, 1.0 µM, and 10 µM or the same amount of DMSO as a control for 2 days. Then, BrdU incorporation was analyzed using a Cell Proliferation Kit (GE Healthcare). At least n=3 tissues were examined.

Spermatogonia of a *sox17::egfp* hyperplasia testis were cultured as described (*Kawasaki et al., 2016*). After 1 month of SSC propagation, SSCs were transferred onto the zebrafish Sertoli cell line ZtA6-12 (*Kurita and Sakai, 2004*) to induce differentiation, and 0.2 µM cycloheximide was added. After 9 days, the SSCs (large cells with a few large nucleoli) and the differentiated spermatogonia (small cells with several small nucleoli) were counted according to their morphology and eGFP expression. n=3 dishes were examined. Since ZtA6-12 cells were maintained in a culture medium containing the 0.22 µm pore size filtered-zebrafish embryo extract at 3 dpf (*Westerfield, 2000*), we did not test the cells for mycoplasma contamination.

## Co-IP

Co-IP was performed with slight modifications as described (*Houwing et al., 2007*). Testes were homogenized with cell lysis buffer M (Wako) containing cOmplete, Mini Protease Inhibitor cocktail (Roche). One IP generally contained 20 µl protein G beads (Protein G HP SpinTrap, Cytiva), three testes, and 20 µg of antibodies against Meioc or Piwil1 (*Supplementary file 4*) in a total volume of 500 µl. The experiments were repeated three times.

For RIP, testes were homogenized with 133 µl of the lysis buffer containing 100 U/ml of SUPERase·In RNase inhibitor (Thermo Fisher Scientific) for 1 mg of testis. One IP contains 30 µl Dynabeads Protein G (Thermo Fisher Scientific), 7.2 µg of anti-Piwil1 antibody, and 250 µl of the lysates. Beads and 1% of the lysates were used for TRIzol (Thermo Fisher Scientific) RNA isolation. RNA was reverse-transcribed using the PrimeScript RT Reagent Kit with gDNA Eraser (Takara) and used for RT-qPCR with TB Green Premix Ex Taq II (Takara) using LightCycler 480 (Roche). The thermal cycling was as follows: initial hold for 2 min at 95°C followed by 60 cycles of 30 s at 95°C, 30 s at 58°C for primer set for unprocessed pre-rRNA (5'ETS-18S rRNA in *Supplementary file 3*; *Heyn et al., 2017*), and 20 s at 72°C. Fold enrichment was calculated with -ddCt by normalization with input using Sigma RIP-qRT-PCR Data Analysis Calculation Shell, associated with the Sigma Imprint RIP kit (http://www.sigmaaldrich.com/life-science/epigenetics/imprint-rna.html). The experiments were repeated three times.

## Mass spectrometry

For mass spectrometry, co-IP using anti-Meioc IgG and normal rat IgG was performed using normal testes and hypertrophied testes as described above. We used Dynabeads Protein G (Thermo Fisher

Scientific) and the anti-Meioc antibody was cross-linked with the beads as described by the manufacturers. The IP was used for mass spectrometry as described (*Ishiguro et al., 2020*). The raw LC-MS/MS data was analyzed against the UniProt Knowledgebase restricted to *Danio rerio* using Proteome Discoverer version 1.4 (Thermo Fisher Scientific) with the Mascot search engine version 2.5 (Matrix Science).

## Construction of expression vectors

For transfection of Meioc, Ythdc2, Piwil1, and HP1α, cDNA fragments encoding zebrafish full-length *meioc*, *ythdc2*, *piwil1*, and *hp1α* were amplified by RT-PCR using primer sets (*Supplementary file 3*). The amplified fragments of *meioc*, *ythdc2*, and *piwil1* were subcloned into a pFLAG-CMV-5a expression vector (Sigma-Aldrich) by using EcoRI and SalI to create a FLAG tag at the C-terminus of the expressed protein. The amplified fragment was assembled via overlap sequence using NEBuilder HiFi DNA Assembly Master Mix (New England BioLabs). Each 500 ng of plasmids was used for transfection to the cells cultured in 35 mm dish.

For transfection of *Escherichia coli* (Rosetta-gami2 [Novagen]), cDNA fragments encoding zebrafish *hp1α* were amplified by RT-PCR using primer sets (*Supplementary file 3*) and cloned into the pET-21a (+) vector (Novagen) using NEBuilder HiFi DNA Assembly Mix (New England BioLabs). Expression of cloned cDNAs was induced with 1 µM isopropyl-β-D-thiogalactopyranoside (Wako).

## Pull-down assay

The expression vectors were transfected into HEK-293A cells using Lipofectamine 3000 Transfection Reagent (Thermo Fisher Scientific). After 48 hr, cells were harvested and used for immunoprecipitation as described above. We used anti-Flag M2 agarose resin (Sigma). The experiments were repeated three times. HEK293A cells were purchased from Thermo Fisher Scientific with confirmation of the absence of mycoplasma contamination.

## Small RNA-seq library preparation and sequencing

Total RNA from six *meioc⁺/ᵐᵒ* and five *meiocᵐᵒ/ᵐᵒ* testis samples at 10 months of age was isolated using TRIzol RNA isolation according to the manufacturer's instructions (Thermo Fisher Scientific). RNAs smaller than 40 nt were isolated using a 15% TBE-Urea polyacrylamide gel (Bio-Rad) and purified with sodium chloride/isopropanol precipitation. NGS library preparation was performed using the NEBNext Multiplex Small RNA Library Prep Set for Illumina (New England BioLabs) as recommended by the manufacturer (protocol version v2.0 8/13), with 14 PCR cycles for library amplification. The PCR-amplified DNA was purified using AMPure XP beads (Beckman Coulter). Size selection of the small RNA library was done on LabChip XT instrument (Perkin Elmer) using a DNA 300 assay kit. The library fractions in the range of 120–161 bp were pooled in equal molar ratio. The resulting 2 nM pool was denatured and diluted to 10 pM with 5% PhiX spike-in DNA and sequenced (single read, 51 cycles, high output mode) on two lanes of HiSeq 2000 system (Illumina).

## Small RNA-seq data analysis

The raw NGS reads in FastQ format were cleaned from partial 3' adapter sequences using Flexbar v.2.4 (*Dodt et al., 2012*) using parameters: `-m 18 -ao 10 -as` *AGATCGGAAGAGCACACGTCTGAACTCCAGTCAC*. Read mapping to the *Danio rerio* reference genome (Zv9/danRer7 build from Illumina iGenomes) was carried out using Bowtie v.0.12.8 (*Langmead et al., 2009*) with parameters: `-n 0 -e 80 -l 18 -y --best --nomaqround`. Reads were assigned to the predicted zebrafish 28S and 18S rRNAs, defined as 4110 nt and 1887 nt sequences from GenBank records CT956064 and BX537263 with highest Megablast homology to the respective rRNAs from *Cyprinus carpio* (GenBank JN628435), as well as to the zebrafish R2 transposon (GenBank AB097126), using Bowtie with perfect match parameters (-v 0 -m 1). Quality assessment of the raw data and length profiling of the mapped reads was performed with FastQC (http://www.bioinformatics.babraham.ac.uk/projects/fastqc). The sequence data have been deposited in the NCBI GEO repository (http://www.ncbi.nlm.nih.gov/geo) under accession number GSE84060.

## Inhibition of RNA polymerases

Fragments of *meiocᵐᵒ/ᵐᵒ* testes were cultured in the spermatogonia proliferation medium without growth factors (*Kawasaki et al., 2012*). Actinomycin D (Wako) at 1 µg/ml, α-amanitin (*Bensaude,*

*2011*) at 10 µg/ml and BMH-21 (Sigma) (*Colis et al., 2014*) at 1 µM were added. After 1 hr incubation at 28°C in 5% $CO_2$/20% $O_2$, testicular samples were fixed and analyzed by immunohistochemistry for Piwil1. The experiments were repeated three times.

### In situ hybridization and immunocytochemistry of isolated spermatogonia

After sorting spermatogonia of *sox17::egfp* wild-type that expresses EGFP at the equivalent stage to *meioc^mo/mo* spermatogonia (*Kawasaki et al., 2012*) using JSAN cell sorter (Bay Bioscience), cells were plated on the CREST-coated glass slides (Matsunami) for 10 min and fixed with 4% paraformaldehyde for 10 min. The cells were treated with 0.5% Triton X-100 in PBS for 5 min. To detect 28S rRNA, the cells were acetylated with 0.025% acetic anhydride in triethanolamine 10 mM for 5 min and followed as described above. After hybridization, antibodies and reagents (*Supplementary file 4*) were used. Images were obtained under an FV1000 confocal microscope (Olympus). The signal strength of 28S rRNA, Meioc, and Piwil1 was quantified using ImageJ/Fiji software (*Schindelin et al., 2012*). The signal intensities of in situ hybridization (28S rRNA) and immunocytochemistry (Meioc and Piwil1) were normalized to the intensities of sense probe and control IgG, respectively.

To quantify the number of Meioc granules per cell, manual focal adjustment and granule counting were performed to ensure that all intracellular granules were detected and counted. In cases where granule number exceeded 50, accurate enumeration became unreliable due to the risk of double counting. Therefore, precise quantification was performed only for cells containing fewer than 50 granules, while those with 50 or more granules were recorded as '50≤'.

### DNA methylation analysis

Because there was no information of the structure of IGS region in zebrafish, we analyzed IGS sequence of known active locus of 45S-S rDNA (Ch5: 831,755–826,807 in GRCz11) (*Locati et al., 2017*) using tandem repeats finder software (*Benson, 1999*). The result was summarized in the schema of *Figure 2—figure supplement 2*. The undifferentiated spermatogonia of *meioc^mo/mo;sox17::egfp* and *sox17::egfp* wild-type were sorted using JSAN cell sorter (Bay Bioscience) and used for bisulfite conversion using MethylEasy Xceed Rapid DNA Bisulfite modification kit (Takara). The tandem repeat region of 45S-S rDNA IGS was amplified by PCR with EpiTaq HS (Takara) using the primer sets (*Supplementary file 3*). The primers were designed using Meth primer software (*Li and Dahiya, 2002*). All PCR products were subcloned into the pCRII vector (Thermo Fisher Scientific) and used for sequencing analysis using online QUMA software (*Kumaki et al., 2008*).

### Comparison of RNA expression levels

After sorting spermatogonia of *vas::egfp; meioc^mo/mo* and *sox17::egfp* wild-type that expresses EGFP at the equivalent stage to *meioc^mo/mo* spermatogonia using JSAN cell sorter (Bay Bioscience), total RNA was extracted and used for RT-qPCR analysis as described in 'Co-IP' section using primer sets (*Supplementary file 3*). Relative gene expression levels were calculated using the comparative Ct method (*Schmittgen and Livak, 2008*) and normalized to the expression of *gapdh*. The experiments were repeated three times.

### Northern blot analysis

Northern blot analysis was performed using DIG northern starter kit (Roche). cDNA fragments of 5'ETS and ITS1 region of 45S-S rDNA (*Locati et al., 2017*) and 7SL were amplified from RT-PCR of testicular RNA using the primers containing T7 and T3 promoter sequence (*Supplementary file 3*) and used for cRNA probe synthesis with T3 RNA polymerase using DIG RNA labeling kit (Roche). Signals of 45S pre-rRNA, pre-rRNA intermediates, and 7SL were detected with a Chemidoc XRS Plus (Bio-Rad) and quantified using quantity tools of ImageLab software version 6.0.1 (Bio-Rad). For the quantification, three wild-type and *meioc^mo/mo* testes were individually used. Signal intensities of 45S pre-rRNA, pre-rRNA intermediates were normalized with signal intensities of 7SL.

### ChIP-qPCR analysis

ChIP assay was performed with slight modifications as described (*Imai et al., 2017*). Ten testes were used for 6 ml of chromatin suspension. Sonication was carried out with a Bioruptor Standard apparatus

(Diagenode) at high power for four series of seven cycles (30 s on, 30 s off). For IP, 1 ml of chromatin suspension was incubated with 20 μl of Dynabeads Protein A (Thermo Fisher Scientific) preincubated with 3 μg of rabbit anti-histone H3 trimethyl K9 (Abcam, ab8898) and Piwil1 antibodies. Quantitative PCR (qPCR) was performed with SYBR Premix Ex Taq II (Takara) using LightCycler 480 (Roche) using primer sets (**Supplementary file 3**). Fold enrichment was calculated with -ddCt by normalization with 10% input sample using Sigma RIP-qRT-PCR Data Analysis Calculation Shell, associated with the Sigma Imprint RIP kit (http://www.sigmaaldrich.com/life-science/epigenetics/imprint-rna.html). The experiments were repeated six times.

## Quantification and statistical analysis

Data were presented as the mean ± standard deviation of at least three independent experiments as indicated in each method and figure legend. Statistical difference between two groups was determined using unpaired Student's t-test when the variance was heterogeneous between the groups, and Welch's t-test was used when the variance was heterogeneous. $p<0.05$ was considered statistically significant. Graphical presentations were made with the R package ggplot2 (**Wickham, 2009**).

## Acknowledgements

We thank C Nüsslein-Volhard for supporting the ENU mutagenesis screening and providing the anti-Ddx4 antibody, K Saito and Y Kato for providing advice, Y Saga for reading the manuscript, Y Yoshida and Y Yamazaki for maintaining the zebrafish stocks, and the IMB Genomics and Bioinformatics Core Facilities. Isolation of the moto mutant was supported by EC Contract LSHG-CT-2003-503496 and meioc[sa13122] was provided by the Zebrafish International Resource Center. This work was supported by JSPS KAKENHI Grant Numbers JP23116709, JP25251034, JP25114003 to NS, JST A-step Grant Numbers JPMJTR204F to NS, the program of the Inter-University Research Network for High Depth Omics, IMEG, Kumamoto University to NS, NIH NICHD (National Institute of Child Health and Human Development) R15HD107594 to KRS. CR was supported by the UMB Initiative for Maximizing Student Development program, NIH award R25 GM076321.

---

## Additional information

### Funding

| Funder | Grant reference number | Author |
| --- | --- | --- |
| Japan Society for the Promotion of Science | JP23116709 | Noriyoshi Sakai |
| Japan Science and Technology Agency | Jst-A step JPMJTR204F | Noriyoshi Sakai |
| National Institutes of Health | The UMB Initiative for Maximizing Student Development program, NIH award R25 GM076321 | Carina Ramos |
| Japan Society for the Promotion of Science | JP25251034 | Noriyoshi Sakai |
| Japan Society for the Promotion of Science | JP25114003 | Noriyoshi Sakai |
| Kumamoto University | Inter-University Research Network for High Depth Omics, IMEG, Kumamoto University | Noriyoshi Sakai |
| National Institute of Child Health and Human Development | R15HD107594 | Kellee R Siegfried |

The funders had no role in study design, data collection and interpretation, or the decision to submit the work for publication.

## Author contributions
Toshihiro Kawasaki, Kellee R Siegfried, Investigation, Writing – original draft, Writing – review and editing; Toshiya Nishimura, Naoki Tani, Emil Karaulanov, Minori Shinya, Kenji Saito, Emily Taylor, René F Ketting, Investigation; Carina Ramos, Funding acquisition, Investigation; Kei-ichiro Ishiguro, Minoru Tanaka, Investigation, Writing – review and editing; Noriyoshi Sakai, Conceptualization, Funding acquisition, Investigation, Writing – original draft, Writing – review and editing

## Author ORCIDs
René F Ketting ⓘ https://orcid.org/0000-0001-6161-5621
Kei-ichiro Ishiguro ⓘ https://orcid.org/0000-0002-7515-1511
Kellee R Siegfried ⓘ https://orcid.org/0000-0002-7951-392X
Noriyoshi Sakai ⓘ https://orcid.org/0000-0001-8403-8281

## Ethics
The use of zebrafish for experimental purposes was approved by the committee on laboratory animal care and use at the National Institute of Genetics (approval identification numbers, 27-12 and 28-13) and the University of Massachusetts Boston Institutional Animal Care and Use Committee (protocol #20120032), and carried out according to the Animal Research Reporting of In Vivo Experiments (ARRIVE) guidelines and to relevant guidelines and regulations.

Reviewer #1 (Public review): https://doi.org/10.7554/eLife.104295.3.sa1
Reviewer #2 (Public review): https://doi.org/10.7554/eLife.104295.3.sa2
Reviewer #3 (Public review): https://doi.org/10.7554/eLife.104295.3.sa3
Author response https://doi.org/10.7554/eLife.104295.3.sa4

# Additional files

## Supplementary files
Supplementary file 1. LC/MS/MS for the immunoprecipitate (IP) of Meioc with lysate of a wild-type testis. Proteins in anti-Meioc IP sample with at least threefold enrichment in normal testes compared with control IgG IP sample, as detected by mass spectrometry. Uniprot IDs are shown in Accession. Proteins were classified with GO terms (http://amigo.geneontology.org). The molecular weight, number of peptides identified by spectrometry, and coverage are provided. Meioc protein is highlighted as red, and known mouse MEIOC partner YTHDC2 is indicated by bold.

Supplementary file 2. LC/MS/MS for the immunoprecipitate (IP) of Meioc with lysate of a hyperplasia testis in which spermatogonial stem cells (SSCs) accumulate. Proteins in anti-Meioc IP sample with at least threefold enrichment in the hyperplasia testis compared with control IgG IP sample, as detected by mass spectrometry. NCBI protein ID or Uniprot ID is shown in Accession. Proteins were classified with GO terms (https://amigo.geneontology.org/amigo). The molecular weight and number of peptides identified by spectrometry, and coverage are provided. Meioc protein is highlighted as red. *: protein recorded in RNA Granule Database as processing body (P-body) protein, **: protein recorded in RNA Granule Database as stress granule protein (Youn JY, Dunham WH, Hong SJ, Knight JDR, Bashkurov M, Chen GI, Bagci H, Rathod B, MacLeod G, Eng SWM, et al. 2018. High-Density Proximity Mapping Reveals the Subcellular Organization of mRNA-Associated Granules and Bodies. Mol Cell 69: 517-532. doi: 10.1016/j.molcel.2017.12.020).

Supplementary file 3. Oligonucleotide primers and sgRNA used in this study. *: Primer set to amplify upstream of rhodopsin locus (Morley RH, Lachani K, Keefe D, Gilchrist MJ, Flicek P, Smith JC, Wardle FC. 2009. A gene regulatory network directed by zebrafish. No tail accounts for its roles in mesoderm formation. Proc Natl Acad Sci 106: 3829–3834. doi: 10.1073/pnas.0808382106).

Supplementary file 4. Antibodies and reagents used in this study.

MDAR checklist

## Data availability
The sequence data have been deposited in the NCBI GEO repository (http://www.ncbi.nlm.nih.gov/geo) under accession number GSE84060.

The following dataset was generated:

| Author(s) | Year | Dataset title | Dataset URL | Database and Identifier |
|---|---|---|---|---|
| Sakai N, Siegfried KR, Karaulanov E, Ketting R | 2025 | Analysis of piRNA production in meioc (moto) mutant zebrafish testis | https://www.ncbi.nlm.nih.gov/geo/query/acc.cgi?acc=GSE84060 | NCBI Gene Expression Omnibus, GSE84060 |

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
