## [Editor Report · eLife Assessment]

This **important** paper describes the regulatory pathway of rRNA synthesis by Meioc-Piwil1 in germ cell differentiation in zebrafish. Using the molecular genetic and cytological approaches, the authors provide **convincing** evidence that Meioc antagonizes Piwil1, which downregulates the 45S pre-rRNA synthesis by heterochromatin formation for spermatocyte differentiation. The results will be of use to researchers in the field of germ cell/meiosis as well as RNA biosynthesis and chromatin.

---

## [Referee Report · Reviewer #1 (Public review)]

Summary:

In this paper Kawasaki et al describe a regulatory role for the PIWI/piRNA pathway in rRNA regulation in Zebrafish. This regulatory role was uncovered through a screen for gonadogenesis defective mutants, which identified a mutation in the meioc gene, a coiled-coil germ granule protein. Loss of this gene leads to redistribution of Piwil1 from germ granules to the nucleolus, resulting in silencing of rRNA transcription.

Strengths:

Most of the experimental data provided in this paper is compelling. It is clear that in the absence of meioc, PiwiL1 translocates in to the nucleolus and results in down regulation of rRNA transcription. the genetic compensation of meioc mutant phenotypes (both organismal and molecular) through reduction in PiwiL1 levels are evidence for a direct role for PiwiL1 in mediating the phenotypes of meioc mutant.

Weaknesses:

Questions remain on the mechanistic details by which PiwiL1 mediated rRNA down regulation, and whether this is a function of Piwi in an unperturbed/wildtype setting. There is certainly some evidence provided in support of the a natural function for piwi in regulating rRNA transcription (figure 5A+5B). However, the de-enrichment of H3K9me3 in the heterozygous (Figure 6F) is very modest and in my opinion not convincingly different relative to the control provided. It is certainly possible that PiwiL1 is regulating levels through cleavage of nascent transcripts. Another aspect I found confounding here is the reduction in rRNA small RNAs in the meioc mutant; I would have assumed that the interaction of PiwiL1 with the rRNA is mediated through small RNAs but the reduction in numbers do not support this model. But perhaps it is simply a redistribution of small RNAs that is occurring. Finally, the ability to reduce PiwiL1 in the nucleolus through polI inhibition with actD and BMH-21 is surprising. What drives the accumulation of PiwiL1 in the nucleolus then if in the meioc mutant there is less transcription anyway?

Despite the weaknesses outlined, overall I find this paper to be solid and valuable, providing evidence for a consistent link between PIWI systems and ribosomal biogenesis. Their results are likely to be of interest to people in the community, and provide tools for further elucidating the reasons for this link.

---

## [Referee Report · Reviewer #2 (Public review)]

Summary:

In this study, the authors report that Meioc is required to upregulate rRNA transcription and promote differentiation of spermatogonial stem cells in zebrafish. The authors show that upregulated protein synthesis is required to support spermatogonial stem cells' differentiation into multi-celled cysts of spermatogonia. Coiled coil protein Meioc is required for this upregulated protein synthesis and for increasing rRNA transcription, such that the Meioc knockout accumulates 1-2 cell spermatogonia and fails to produce cysts with more than 8 spermatogonia. The Meioc knockout exhibits continued transcriptional repression of rDNA. Meioc interacts with and sequesters Piwil1 to the cytoplasm. Loss of Meioc increases Piwil1 localization to the nucleolus, where Piwil1 interacts with transcriptional silencers that repress rRNA transcription.

Strengths:

This is fundamental study that expands our understanding of how ribosome biogenesis contributes to differentiation and demonstrates that zebrafish Meioc plays a role in this process during spermatogenesis. This work also expands our evolutionary understanding of Meioc and Ythdc2's molecular roles in germline differentiation. In mouse, the Meioc knockout phenocopies the Ythdc2 knockout, and studies thus far have indicated that Meioc and Ythdc2 act together to regulate germline differentiation. Here, in zebrafish, Meioc has acquired a Ythdc2-independent function. This study also identifies a new role for Piwil1 in directing transcriptional silencing of rDNA.

Comments on revisions:

Major and minor concerns were addressed in the revision.

---

## [Referee Report · Reviewer #3 (Public review)]

Summary:

The paper describes the molecular pathway to regulate germ cell differentiation in zebrafish through ribosomal RNA biogenesis. Meioc sequesters Piwil1, a Piwi homolog, which suppresses the transcription of the 45S pre-rDNA by the formation of heterochromatin, to the perinuclear bodies.

Strong points:

The authors nicely provided the molecular evidence on the antagonism of Meioc to Piwil1 in the rRNA synthesis, which supported by the genetic evidence that the inability of the meioc mutant to enter meiosis is suppressed by the piwil1 heterozygosity. The authors nicely address my previous points.

Weak points:

Although the authors made an effort to revise the text. However, there are still some points that the authors need to check their text. Some of them are shown in "Minor points" below. I am sorry that some of them should have been pointed in my previous review.

---

## [Author Response]

The following is the authors’ response to the original reviews

**Public Reviews:**

**Reviewer #1 (Public review):**
Summary:In this paper Kawasaki et al describe a regulatory role for the PIWI/piRNA pathway in rRNA regulation in Zebrafish. This regulatory role was uncovered through a screen for gonadogenesis defective mutants, which identified a mutation in the meioc gene, a coiled-coil germ granule protein. Loss of this gene leads to redistribution of Piwil1 from germ granules to the nucleolus, resulting in silencing of rRNA transcription.Strengths:Most of the experimental data provided in this paper is compelling. It is clear that in the absence of meioc, PiwiL1 translocates in to the nucleolus and results in down regulation of rRNA transcription. the genetic compensation of meioc mutant phenotypes (both organismal and molecular) through reduction in PiwiL1 levels are evidence for a direct role for PiwiL1 in mediating the phenotypes of meioc mutant.Weaknesses:Questions remain on the mechanistic details by which PiwiL1 mediated rRNA down regulation, and whether this is a function of Piwi in an unperturbed/wildtype setting. There is certainly some evidence provided in support of the natural function for piwi in regulating rRNA transcription (figure 5A+5B). However, the de-enrichment of H3K9me3 in the heterozygous (Figure 6F) is very modest and in my opinion not convincingly different relative to the control provided. It is certainly possible that PiwiL1 is regulating levels through cleavage of nascent transcripts. Another aspect I found confounding here is the reduction in rRNA small RNAs in the meioc mutant; I would have assumed that the interaction of PiwiL1 with the rRNA is mediated through small RNAs but the reduction in numbers do not support this model. But perhaps it is simply a redistribution of small RNAs that is occurring. Finally, the ability to reduce PiwiL1 in the nucleolus through polI inhibition with actD and BMH-21 is surprising. What drives the accumulation of PiwiL1 in the nucleolus then if in the meioc mutant there is less transcription anyway?Despite the weaknesses outlined, overall I find this paper to be solid and valuable, providing evidence for a consistent link between PIWI systems and ribosomal biogenesis. Their results are likely to be of interest to people in the community, and provide tools for further elucidating the reasons for this link.

The amount of cytoplasmic rRNA in *piwi+/-* was increased by 26% on average (figure 5A+5B), the amount of ChiP-qPCR of H3K9 was decreased by about 26% (Figure 6F), and ChiP-qPCR of Piwil1 was decreased by 35% (Figure 6G), so we don't think there is a big discrepancy. On the other hand, the amount of ChiP-qPCR of H3K9 in *meiocmo/mo* was increased by about 130% (Figure 6F), while ChiP-qPCR of Piwil1 was increased by 50%, so there may be a mechanism for H3K9 regulation of Meioc that is not mediated by Piwil1. As for what drives the accumulation of Piwil1 in the nucleolus, although we have found that Piwil1 has affinity for rRNA (Fig. 6A), we do not know what recruits it. Significant increases in the 18-35nt small RNA of 18S, 28S rRNAs and R2 were not detected in *meiocmo/mo* testes enriched for 1-8 cell spermatogonia, compared with *meioc+/mo* testes. The nucleolar localization of Piwil1 has revealed in this study, which will be a new topic for future research.

**Reviewer #2 (Public review):**
Summary:In this study, the authors report that Meioc is required to upregulate rRNA transcription and promote differentiation of spermatogonial stem cells in zebrafish. The authors show that upregulated protein synthesis is required to support spermatogonial stem cells' differentiation into multi-celled cysts of spermatogonia. Coiled coil protein Meioc is required for this upregulated protein synthesis and for increasing rRNA transcription, such that the Meioc knockout accumulates 1-2 cell spermatogonia and fails to produce cysts with more than 8 spermatogonia. The Meioc knockout exhibits continued transcriptional repression of rDNA. Meioc interacts with and sequesters Piwil1 to the cytoplasm. Loss of Meioc increases Piwil1 localization to the nucleolus, where Piwil1 interacts with transcriptional silencers that repress rRNA transcription.Strengths:This is a fundamental study that expands our understanding of how ribosome biogenesis contributes to differentiation and demonstrates that zebrafish Meioc plays a role in this process during spermatogenesis. This work also expands our evolutionary understanding of Meioc and Ythdc2's molecular roles in germline differentiation. In mouse, the Meioc knockout phenocopies the Ythdc2 knockout, and studies thus far have indicated that Meioc and Ythdc2 act together to regulate germline differentiation. Here, in zebrafish, Meioc has acquired a Ythdc2-independent function. This study also identifies a new role for Piwil1 in directing transcriptional silencing of rDNA.Weaknesses:There are limited details on the stem cell-enriched hyperplastic testes used as a tool for mass spec experiments, and additional information is needed to fully evaluate the mass spec results. What mutation do these testes carry? Does this protein interact with Meioc in the wildtype testes? How could this mutation affect the results from the Meioc immunoprecipitation?

Stem cell-enriched hyperplastic testes came from wild-type adult *sox17::GFP* transgenic zebrafish. Sperm were found in these hyperplastic testes, and when stem cells were transplanted, they self-renewed and differentiated into sperm. It is not known if the hyperplasias develop due to a genetic variant in the line. We added the following comment in L201-204.

“The SSC-enriched hyperplastic testes, which are occasionally found in adult wildtype zebrafish, contain cells at all stages of spermatogenesis. Hyperplasia-derived SSCs self-renewed and differentiated in transplants of aggregates mixed with normal testicular cells.”

**Reviewer #3 (Public review):**
Summary:The paper describes the molecular pathway to regulate germ cell differentiation in zebrafish through ribosomal RNA biogenesis. Meioc sequesters Piwil1, a Piwi homolog, which suppresses the transcription of the 45S pre-rDNA by the formation of heterochromatin, to the perinuclear bodies. The key results are solid and useful to researchers in the field of germ cell/meiosis as well as RNA biosynthesis and chromatin.Strengths:The authors nicely provided the molecular evidence on the antagonism of Meioc to Piwil1 in the rRNA synthesis, which supported by the genetic evidence that the inability of the meioc mutant to enter meiosis is suppressed by the piwil1 heterozygosity.Weaknesses:(1) Although the paper provides very convincing evidence for the authors' claim, the scientific contents are poorly written and incorrectly described. As a result, it is hard to read the text. Checking by scientific experts would be highly recommended. For example, on line 38, "the global translation activity is generally [inhibited]", is incorrect and, rather, a sentence like "the activity is lowered relative to other cells" is more appropriate here. See minor points for more examples.

Thank you for pointing that out. I corrected the parts pointed out.

(2) In some figures, it is hard for readers outside of zebrafish meiosis to evaluate the results without more explanation and drawing.

We refined Figure 1A and added explanation about SSC, *sox17::egfp* positive cells, and the SSC-enriched hyperplastic testis in L155-158.

(3) Figure 1E, F, cycloheximide experiments: Please mention the toxicity of the concentration of the drug in cell proliferation and viability.

When testicular tissue culture was performed at 0.1, 1, 10, 100, 250, and 500mM, abnormal strong OP-puro signals including nuclei were found in cells at 10mM or more. We added the results in the Supplemental Figure S2G. In addition, at 1mM, growth was perturbed in fast-growing 32≤-cell cysts of spermatogonia, but not in 1-4-cell spermatogonia, as described in L127-130.

**Recommendations for the authors:**

**Reviewer #1 (Recommendations for the authors):**
I don't have any recommendations for improvement. While I have outlined some of the weaknesses of the paper above. I don't see addressing these questions as pertinent for publication of this paper.
**Reviewer #2 (Recommendations for the authors):**
(1) The manuscript uses the terms 1-2 cell spermatogonia, GSC, and SSC throughout the figures and text. For example, 1-2 cell spermatogonia is used in Figure 1C, GSC is used in Figure 1F, and SSC is used in Figure 1 legend. The use of all three terms without definitions as to how they each relate with one another is confusing, particularly to those outside the zebrafish spermatogenesis field. It would be best to only use one term if the three terms are used interchangeably or to define each term if they represent different populations.

GSC is a writing mistake. In this study, *sox17*-positive cells, which have been confirmed to self-renew and differentiate (Kawasaki et al., 2016), are considered SSCs. On the other hand, a comparison of *meioc* and *ythdc2* mutants revealed differences in the composition of each cyst, so we describe the number of cysts confirmed. We added new data that 1-2 cell spermatogonia are *sox17*-positive in Supplemental Figure S3 (L157-158).

(2) Figure 1B: What does the "SC" label represent in these figure panels?

We added the explanation in the Figure legend.

(3) Fig 7B and S7B show incongruent results, and the text implies that Fig S7B data better reflects in vivo biology. It is not clear how the authors interpret the different results between 7B and S7B.

Thank you for pointing that out. Fig 7A and 7B were obtained by isolating sox17-positive cells. Because it was difficult to detect nucleoli in the isolated cells, probably due to the isolation procedure, we added S7B, which was analyzed in sectioned tissues. As this reviewer pointed out, S7B reflects the in vivo state better, so we changed S7B to 7B and 7B to S7B.

**Reviewer #3 (Recommendations for the authors):**
Minor points:(1) For general readers, it is nice to add a scheme of zebrafish spermatogenesis (lines 77-78) together with Figure 1A.

As mentioned above, we refined Figure 1A.

(2) Line 28, silence: the word "silence" is too strong here since rDNA is transcribed in some levels to ensure the cell survival.

Thank you for your comment. We changed "silence" to "maintain low levels."

(3) Line 60, YTDHC2: Please explain more about what protein YTDHC2 is.

We added a description of Ythdc2 in the introduction.

(4) Line 69, Piwil1: Please explain more about what protein Piwil1 is.

We added a description of Piwil1 in the introduction.

(5) Figure 1B, sperm: Please show clearly which sperms are in this figure using arrows etc.

We represented sperm using arrowheads in Fig 1B.

(6) Figure 1C, SC: Please show what SC is in the legend.

We added the explanation in the Figure legend.

(7) Line 83, meiotic makers: should be "meiotic prophase I makers".

Thank you for pointing out the inaccurate expression description. We revised it.

(8) Line 84, phosphor-histone H3: Should be "histone H3 phospho-S10 "

We revised it.

(9) Figure S1A, PH3: Please add PH3 is "histone H3 phospho-S10 ".

We revised it.

(10) Figure S1A, moto+/-: this heterozygous mutant showed an increased apoptosis. If so, please mention this in the text. If not, please remove the data.

Thank you for pointing that out. The heterozygous mutant did not increase apoptosis, so we removed the data.

(11) Line 88, no females developed: This means all males in the mutant. If so, what Figure S1B shows? These cells are spermatocytes? No "oocytes" developed is correct here?

All *meiocmo/mo* zebrafish were males, and the *meiocmo/mo* cells in Fig. S1B are spermatogonia. No spermatocytes or oocytes were observed. To show this, we added "no oocytes" in L90.

(12) Line 89, initial stages: What do the initial stages mean here? Please explain.

The “initial stages” was changed to the pachytene stage.

(13) Figure S1C: mouse Meioc rectangle lacks a right portion of it. Please explain two mutations encode a truncated protein in the main text.

I apologize. It seems that the portion was missing during the preparation of the manuscript. We corrected it. In addition, we added a description of the protein truncation in L100-101.

(14) Line 99: What "GRCz11" is.

GRCz11 refers to the version of the zebrafish reference genome assembly. We added this.

(15) Figure S2A: Dotted lines are cysts. If so, please mention it in the legend.

We corrected the figure legend.

(16) Figure S2B and C:, B1-4, C1-7: Rather use spermatogonia etc as a caption here.

We corrected the figure and figure legend.

(17) Line 113, hereafter, wildtype: Should be "wild type" or "wild-type".

We corrected them.

(18) Figure 1C: Please indicate what dotted lines mean here.

We added “Dotted lines; 1-2 cell spermatogonia.”

(19) Line 113, de novo: Please italicize it.

We corrected it.

(20) Line 113-116: Figure 1D shows two populations in the protein synthesis (low and high) in the 1-2-cell stage. Please mention this in the text.

We added mention of two population.

(21) Line 121, in vitro: Please italicize it.

We corrected it.

(22) Line 138-139, Figure 2A: Please indicate two populations in the rRNA concentrations (low and high) in the 1-2-cell stage. How much % of each cell is?

We added mention of two population and % of each cell.

(23) Figure 2B, cytes: Please explain the rRNA expression in spermatocytes (cytes) in the text.

The decrease in rRNA signal intensity in spermatocytes was added.

(24) Figure 2A, lines 147, low signals: Figure 2A did not show big differences between wild type and the mutant. What did the authors mean here? Lower levels of rRNAs in the mutant than in wild type. If so, please write the text in that way.

We think that it is important to note that we were unable to find cells with upregulated rRNA signals, and therefore changed to “could not find cells with high signals of rRNAs and Rpl15 in *meiocmo/mo* spermatogonia”.

(25) Figure 2E: Please add a schematic figure of a copy of rDNA locus such as Fig. S3A right.

We added a schema of rDNA locus and primer sites such as Figure S3A right (now Figure 2F) in Figure 2E.

(26) Figure S3A: This Figure should be in the main Figure. The quantification of Northern blots should be shown as a graph with statistical analysis.

We added the quantification and transfer to the main Figure (Figure 2F).

(27) Figure 4A: Please show single-color images (red or green) with merged ones.

We added single-color images in the Figure 4A.

(28) Line 198, Piwil1: Please explain what Piwil1 is briefly.

We are sorry, but we could not quite understand the meaning of this comment. To show that Piwil1 is located in the nucleolus, we indicated it as (Figure 4A, arrowhead) in L209.

(29) Line 198, Ddx4-positive: What is "Ddx4-positive"? Explain it for readers.

Ddx4 is a marker for germinal granules, and the description was changed to reflect this.

(30) Line 209, Fig. S4D-G: Please mention the method of the detection of piRNA briefly.

We have described that we have sequenced small RNAs of 18-35 nt. Accordingly, we changed the term piRNA to small RNA.

(31) Line 217: Please mention piwil1 homozygous mutant are inviable.

We added that *piwil1-/-* are viable in L231.